 **eLIFE**

# Spatial self-organization favors heterotypic cooperation over cheating

**Babak Momeni\*, Adam James Waite, Wenying Shou\***

Division of Basic Sciences, Fred Hutchinson Cancer Research Center, Seattle, United States

**Abstract** Heterotypic cooperation—two populations exchanging distinct benefits that are costly to produce—is widespread. Cheaters, exploiting benefits while evading contribution, can undermine cooperation. Two mechanisms can stabilize heterotypic cooperation. In 'partner choice', cooperators recognize and choose cooperating over cheating partners; in 'partner fidelity feedback', fitness-feedback from repeated interactions ensures that aiding your partner helps yourself. How might a spatial environment, which facilitates repeated interactions, promote fitness-feedback? We examined this process through mathematical models and engineered *Saccharomyces cerevisiae* strains incapable of recognition. Here, cooperators and their heterotypic cooperative partners (partners) exchanged distinct essential metabolites. Cheaters exploited partner-produced metabolites without reciprocating, and were competitively superior to cooperators. Despite initially random spatial distributions, cooperators gained more partner neighbors than cheaters did. The less a cheater contributed, the more it was excluded and disfavored. This self-organization, driven by asymmetric fitness effects of cooperators and cheaters on partners during cell growth into open space, achieves assortment.

**\*For correspondence:**
bmomeni@fhcrc.org (BM);
wenying.shou@gmail.com (WS)

**Competing interests:** The authors declare that no competing interests exist.

**Reviewing editor**: Diethard Tautz, Max Planck Institute for Evolutionary Biology, Germany

## Introduction

Cooperation, providing a benefit available to others at a cost to self, has been postulated to drive major transitions in evolution (*Maynard Smith and Szathmary, 1998*). Cooperation may take place between similar individuals contributing and sharing identical benefits (homotypic cooperation) or between two populations exchanging distinct benefits such as in some forms of mutualism (heterotypic cooperation). Both homotypic and heterotypic cooperation are vulnerable to cheaters (*Turner and Chao, 1999*; *Strassmann et al., 2000*; *Bronstein, 2001*; *Rainey and Rainey, 2003*; *Travisano and Velicer, 2004*). Cheaters exploit cooperative benefits without contributing their fair share and are therefore competitively superior to their cooperating counterparts. How might cooperation avoid being taken over by cheaters? The answer lies in 'positive assortment' (*Fletcher and Doebeli, 2009*), in which benefit-supplying individuals interact more with other benefit-supplying individuals than with cheaters.

In homotypic cooperation that involves genetic relatives, positive assortment can be realized through 'kin discrimination', which is based on the active recognition and preferential treatment of more closely related individuals over distantly related ones (*Sachs et al., 2004*). Positive assortment can also be realized through 'kin fidelity' (*Sachs et al., 2004*). For example, restricted migration in a spatial environment causes homotypic cooperators and cheaters to cluster with their respective progeny. This clustering allows cooperators to preferentially interact with each other (*Figure 1A*, top). Both mechanisms of positive assortment can favor cooperation (*Hamilton, 1964a*; *Hamilton, 1964b*; *Maynard Smith, 1964*; *Chao and Levin, 1981*; *Nowak and May, 1992*; *Fletcher and Doebeli, 2006*; *Kerr et al., 2006*; *MacLean and Gudelj, 2006*; *West et al., 2006*; *Lion and Baalen, 2008*; *Wild et al., 2009*; *West and Gardner, 2010*). A spatial environment may also impede homotypic cooperation by intensifying competition among cooperators (*Taylor, 1992*; *Wilson et al., 1992*;

**eLife digest** Cooperation between individuals of the same species, and also between different species, is known to be important in evolution. Large fish, for example, rely on small cleaner fish to remove parasites, while the small fish benefit from the nutrients in these parasites. However, cooperation can be undermined by other individuals or species who "cheat" by taking advantage of those who cooperate, without providing any benefits in return. For example, some cleaner fish cheat by biting off healthy tissue from their host, in addition to parasites.

Genetically-related individuals who cooperate by sharing identical benefits can combat cheaters by giving preferential treatment to their relatives (a process known as kin discrimination) or by staying close to the relatives to form clusters (kin fidelity). However, two genetically-unrelated populations that mutually cooperate by sharing different benefits cannot employ these methods to overcome cheaters. Instead they rely on either partner choice or partner fidelity feedback.

Partner choice – the approach adopted by cleaner fish and their hosts – relies on one population recognizing a signal from the other population and responding accordingly: for example, large fish observe cleaner fish and approach those that cooperate with their current host and avoid those that cheat. Partner fidelity feedback, on the other hand, relies on repeated interactions between the two populations providing an advantage in terms of evolutionary fitness to both: for example, organelles called mitochondria and chloroplasts live inside cells, helping the cells to harvest energy and providing energy for themselves and the host cells in the process. In some cases – such as the cooperation between figs and fig wasps, or between certain plants and the bacteria that fix nitrogen in their roots – researchers cannot agree if the populations are relying on partner choice or partner fidelity feedback.

Now Momeni et al. have used a combination of experiments on yeast and mathematical modeling to explore partner fidelity feedback in greater detail. They started by using genetic engineering techniques to produce two species of yeast that mutually cooperate, each providing a metabolite that is essential to the other, but are not able to recognize each other: this means that these populations cannot rely on partner choice to combat cheaters. Momeni et al. then observed how these two species interacted with each other and a third species of yeast that cheated by consuming one of the metabolites without releasing any metabolite of its own.

Momeni et al. found that as long as there was space for the yeast cells to grow into, the two species that cooperated self-organized into mixed clusters, with the cheating species being excluded from these clusters. The self-organization was driven by a positive feedback loop involving the two species that cooperated, with each species helping to increase the fitness of the other. The results of Momeni et al. demonstrate that it is possible for two genetically unrelated populations to cooperate and combat cheaters without the use of partner choice.

*West et al., 2002*) and in certain cases, by potentially encouraging cheating strategies (*Hauert and Doebeli, 2004*).

Heterotypic cooperation can occur between populations that are genetically related but phenotypically differentiated, such as between different cell types in a multicellular organism. Alternatively, heterotypic cooperation can involve two genetically unrelated populations (e.g., species) exchanging distinct benefits that are costly to produce. For example, legume plants supply organic carbon and other essential nutrients to rhizobia, and rhizobia reciprocate with fixed nitrogen (*Udvardi and Poole, 2013*). Positive assortment in heterotypic cooperation can be achieved through 'partner choice' and 'partner fidelity feedback' (*Bull and Rice, 1991*; *Sachs et al., 2004*; *Foster and Wenseleers, 2006*; *Weyl et al., 2010*). In partner choice, active mechanisms (disruptable through for example mutational or pharmacological means) enable an individual to differentially reward cooperative instead of non-cooperative partners based on a signal. Thus, discrimination mediated by partner choice can occur in advance of exploitation (*Bull and Rice, 1991*). For instance, in the mutualism between client fish and cleaner fish in which cleaner fish obtain food from removing client parasites, client fish recognize and avoid cheating cleaners that also bite healthy tissues (*Bshary, 2002*). In partner fidelity feedback, fitness-feedback from repeated interactions ensures that aiding the partner helps self. Examples of partner fidelity feedback can be found in mutualism between hosts and their vertically transmitted

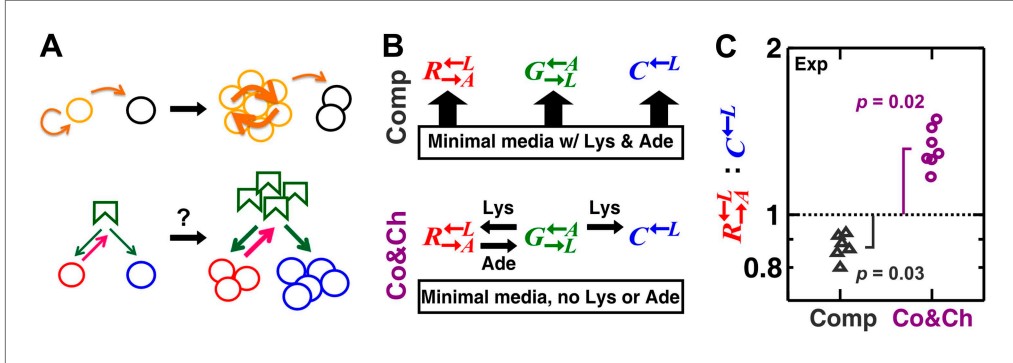

**Figure 1**. A spatial environment favors heterotypic cooperation over cheating. (**A**) Top: clustering with self-type can favor homotypic cooperators (yellow) over cheaters (black). Bottom: clustering with self-type should not favor heterotypic cooperation since cooperator clusters (red) and competitively superior cheater clusters (blue) should have equivalent access to the heterotypic cooperative partner (green). (**B**) We engineered three yeast strains: a red-fluorescent $R^{\leftarrow L}_{\rightarrow A}$ strain requiring lysine and releasing adenine; a green-fluorescent $G^{\leftarrow A}_{\rightarrow L}$ strain requiring adenine and releasing lysine; and a cyan-fluorescent $C^{\leftarrow L}$ strain requiring lysine and not releasing adenine. The three strains purely competed ('Comp') or additionally cooperated and cheated ('Co&Ch'), depending on whether the medium contained or lacked adenine ('Ade') and lysine ('Lys'), respectively. (**C**) In competitive communities, $R^{\leftarrow L}_{\rightarrow A}:C^{\leftarrow L}$ dropped below the initial value of 1 (dotted line) during community growth due to the fitness advantage of $C^{\leftarrow L}$ over $R^{\leftarrow L}_{\rightarrow A}$. In contrast, $R^{\leftarrow L}_{\rightarrow A}:C^{\leftarrow L}$ rose above 1 when the strains engaged in cooperation and cheating. Population ratios in experiments were measured using flow cytometry. All communities started from an initial density of 3000 total cells/mm² with the three strains at a 1:1:1 ratio on top of an agarose column (*Figure 1—figure supplement 1A*) and were analyzed after approximately six to eight generations (*Figure 1—figure supplement 2*). p Values are from the Wilcoxon signed rank test, comparing the median with 1.

The following figure supplements are available for figure 1:

**Figure supplement 1**. Community setup.

**Figure supplement 2**. In experimental spatial communities, cooperators $R^{\leftarrow L}_{\rightarrow A}$ are favored over cheaters $C^{\leftarrow L}$.

symbionts, for example, between eukaryotes and their endosymbioic mitochondria and chloroplasts (*Sachs et al., 2011*).

Natural heterotypic cooperative systems often benefit from a combination of partner choice and partner fidelity feedback. Often, for a particular system, it is challenging to determine which mechanism is mainly responsible for fending off cheaters. For instance, the mutualism between fig and fig wasp and between legume and rhizobia have been thought to employ partner choice by some investigators (*Kiers et al., 2003*; *Sachs et al., 2004*; *Foster and Wenseleers, 2006*; *Jandér and Herre, 2010*) and partner fidelity feedback by others (*Weyl et al., 2010*).

In this study, using engineered yeast strains and mathematical models devoid of possibilities for partner choice, we examined how through partner fidelity feedback heterotypic cooperation between microbes may be protected against cheaters. Spatial environment, which facilitates repeated interactions between neighboring individuals, has been shown to promote heterotypic cooperation (*Boucher et al., 1982*; *Doebeli and Knowlton, 1998*; *Yamamura et al., 2004*; *West et al., 2007*; *Harcombe, 2010*; *Mitri et al., 2011*). However, the mechanism for how partner fidelity feedback unfolds in a spatial environment is not well understood. Specifically, how might spatial correlation in the tendency to contribute arise between genetically unrelated heterotypic cooperators when such correlation was initially absent (*Frank, 1994*)? If population viscosity was the sole driving force, then clusters of cooperators and clusters of competitively superior cheaters would be expected to have equivalent access to clusters of heterotypic cooperative partners. This would seem to favor cheaters (*Figure 1A*, bottom). Instead, we show that in a spatial environment, asymmetric fitness effects of cooperators and cheaters on partners during cell growth into open space drives assortment. This emergence of non-random patterns from initially random spatial distributions, known as self-organization, automatically grants cooperators instead of cheaters more access to heterotypic cooperative partners, disfavoring cheaters. Thus, partner fidelity feedback through self-organization excludes cheaters without evolving recognition mechanisms.

## Results

### Environment-dependent engineered heterotypic cooperation and cheating

To examine how partner fidelity feedback unfolds in a spatial environment, we started with an engineered experimental system incapable of partner recognition (*Shou et al., 2007*; *Waite and Shou, 2012*). This system consisted of three reproductively isolated *Saccharomyces cerevisiae* strains: a green-fluorescent strain requiring adenine and releasing lysine ($G_{\to L}^{\leftarrow A}$), a red-fluorescent strain requiring lysine and releasing adenine ($R_{\to A}^{\leftarrow L}$), and a cyan-fluorescent strain requiring lysine and not releasing adenine ($C^{\leftarrow L}$). Release of lysine or adenine was caused by metabolite overproduction due to a mutation that made the first enzyme of the biosynthetic pathway (Lys21 and Ade4, respectively) insensitive to end-product inhibition (*Armitt and Woods, 1970*; *Feller et al., 1999*). $C^{\leftarrow L}$ still produced adenine for itself at the wild-type level, but without the overproduction mutation, the adenine produced by $C^{\leftarrow L}$ was not sufficient to support the growth of $G_{\to L}^{\leftarrow A}$ (Figure supplement 6 in *Shou et al., 2007*).

These strains engaged in different types of interactions depending on the environment. In minimal medium supplemented with abundant adenine and lysine, they competed for nutrients required by all three strains (e.g., glucose and nitrogen) and limited space (*Figure 1B*, 'Comp'). In minimal medium without supplements, in addition to competition, $G_{\to L}^{\leftarrow A}$ and $R_{\to A}^{\leftarrow L}$ exchanged essential metabolites lysine and adenine (*Shou et al., 2007*), while $C^{\leftarrow L}$ consumed lysine without releasing adenine. Not overproducing adenine, $C^{\leftarrow L}$ showed a ~2% growth rate advantage over $R_{\to A}^{\leftarrow L}$ when lysine was abundant (competition assay in Figure 1 of *Waite and Shou, 2012*). In the absence of lysine, there was no significant difference in the death rates of $R_{\to A}^{\leftarrow L}$ and $C^{\leftarrow L}$ (Figure S1 in *Waite and Shou, 2012*). Finally, in media lacking lysine and adenine, binary cocultures of $R_{\to A}^{\leftarrow L}$ and $G_{\to L}^{\leftarrow A}$ could grow from low to high cell densities (Figure 1 in *Shou et al., 2007*), whereas cocultures of $C^{\leftarrow L}$ and $G_{\to L}^{\leftarrow A}$ failed to grow (Figure supplement 6 in *Shou et al., 2007*). These results collectively suggest that $C^{\leftarrow L}$ acts as a cheater variant of $R_{\to A}^{\leftarrow L}$ (*Waite and Shou, 2012*). In other words, in the absence of supplements, cooperator $R_{\to A}^{\leftarrow L}$ and the competitively superior cheater $C^{\leftarrow L}$ competed for the lysine supplied by the heterotypic cooperative partner (partner) $G_{\to L}^{\leftarrow A}$ (*Figure 1B*, 'Co&Ch'). Cooperator $R_{\to A}^{\leftarrow L}$ 'reciprocated' by releasing adenine, which is essential for partner $G_{\to L}^{\leftarrow A}$, but cheater $C^{\leftarrow L}$ did not release adenine.

### A spatial environment can stabilize heterotypic cooperation against cheaters

We first verified that a spatial environment could stabilize heterotypic cooperation against a competitively superior cheater in our system. We initiated experimental communities from randomly distributed equal proportions of the three cell populations on agarose pads (*Figure 1—figure supplement 1A*). During pure competition in the presence of supplemented adenine and lysine, the $R_{\to A}^{\leftarrow L}{:}C^{\leftarrow L}$ ratio dropped below the original value of 1 after approximately six to eight generations (*Figure 1C*), consistent with the known 2% growth advantage of $C^{\leftarrow L}$ over $R_{\to A}^{\leftarrow L}$ (*Waite and Shou, 2012*). In contrast, during cooperation and cheating in the absence of adenine and lysine supplements, cooperating $R_{\to A}^{\leftarrow L}$ was favored over cheating $C^{\leftarrow L}$ (*Figure 1C*, time course in *Figure 1—figure supplement 2*).

If the spatial aspect of the environment is disrupted, either by periodically mixing a community or by growing it as a well-mixed liquid coculture, partner fidelity feedback should not operate and cheaters are expected to be favored over cooperators. However, the ratio of cooperators $R_{\to A}^{\leftarrow L}$ to cheaters $C^{\leftarrow L}$ in periodically mixed replicate communities varied dramatically (*Figure 2—figure supplement 1*). This was because the lysine-limited environment strongly selected for adaptive mutants in $C^{\leftarrow L}$ and $R_{\to A}^{\leftarrow L}$ (*Waite and Shou, 2012*). Thus, $C^{\leftarrow L}$ and $R_{\to A}^{\leftarrow L}$ engaged in an 'adaptive race' (*Waite and Shou, 2012*) akin to clonal interference: if $C^{\leftarrow L}$ had the best mutation to grow under lysine limitation, then the coculture was quickly destroyed by cheaters; if $R_{\to A}^{\leftarrow L}$ had the best adaptive mutation, then the coculture quickly purged cheaters. As a result, $C^{\leftarrow L}$ outcompeted $R_{\to A}^{\leftarrow L}$ or $R_{\to A}^{\leftarrow L}$ outcompeted $C^{\leftarrow L}$ depending on which population had a better mutation, not because of social interactions. This kind of phenomenon has also been observed for non-engineered cooperating and cheating microbes (*Morgan et al., 2012*). We chose two evolving cocultures in which the $C^{\leftarrow L}$ populations were increasing in frequency, When we grew these two cocultures in well-mixed and in periodically perturbed spatial environments, $C^{\leftarrow L}$ was favored (*Figure 2—figure supplement 2*; 'Materials and methods'). However, in a spatial environment, $R_{\to A}^{\leftarrow L}$ was favored (*Figure 2—figure supplement 2*), consistent with previous experiments on a different microbial system (*Harcombe, 2010*). Our experiments involved non-clonal and non-isogenic

populations. In nature, cheaters can be of different species (such as the non-pollinating wasps of fig) and therefore not isogenic with their cooperating counterpart. Additionally, upon environmental stresses, originally isogenic cooperators and cheaters can quickly acquire different mutations and become nonisogenic (*Morgan et al., 2012*; *Waite and Shou, 2012*). Regardless, we resorted to mathematical models to eliminate the confounding influence of adaptive mutations.

We extended a three-dimensional individual-based model of community growth (previously described as the 'diffusion model' in *Momeni et al., 2013*) to include cooperators, cheaters, and heterotypic partners. The main assumptions of this model were: (1) that the growth of individual cells depended on consumption of the limiting metabolite, and the consumption rate in turn depended on the local concentration of the limiting metabolite according to Michaelis–Menten kinetics; (2) that spatial distribution of metabolites was governed by release, diffusion, and consumption; and (3) that cells rearranged when necessary to accommodate new cells as per experimental observations ('Materials and methods'). Parameters of this model (such as the rates of growth, death, and metabolite consumption and release) were experimentally determined (*Momeni et al., 2013*), mostly through characterizing properties of monocultures (*Figure 2—source data 1*). $C^{\leftarrow L}$ was assumed to have a constant intrinsic fitness advantage over $R^{\leftarrow L}_{\to A}$ at all lysine concentrations, as modeled by a higher maximum uptake rate ($v_{m,C} > v_{m,R}$ in 'The diffusion model' in 'Materials and methods'). In the absence of adaptations, simulation results confirmed that a spatial environment favored cooperators over cheaters and that disrupting the spatial aspect of the environment gave cheaters an advantage over cooperators (*Figure 2*).

During spatial growth of the yeast community, ancestral $R^{\leftarrow L}_{\to A}$ and $C^{\leftarrow L}$ should also engage in an adaptive race to adapt to the lysine-limited environment. However, unlike in the liquid environment, mutants in $R^{\leftarrow L}_{\to A}$ and $C^{\leftarrow L}$ were spatially restricted to the neighborhood of their origins, and thus these mutants could not sweep through the entire community. Consequently, population dynamics from different replicates were highly reproducible (*Figure 1—figure supplement 2*). Thus, we used ancestral $R^{\leftarrow L}_{\to A}$ and $C^{\leftarrow L}$ for all other experiments.

## Differential spatial association with partner favors heterotypic cooperation over cheating

How might a spatial environment promote heterotypic cooperation? The relative positioning of cells in a community, the 'spatial pattern' of a community, can develop differently based on the fitness effects of cell–cell interactions (*Momeni et al., 2013*). Due to the inability of confocal or two-photon microscopy to yield three-dimensional patterns of cells in yeast communities, we only had access to the top-views (xy) and vertical cross-sections (z), with the latter being obtained through cryosectioning (*Momeni and Shou, 2012*). In vertical cross-sections, we have previously shown that in the absence of adenine and lysine supplements, $R^{\leftarrow L}_{\to A}$ and $G^{\leftarrow A}_{\to L}$ as a strongly cooperative pair (i.e., there is a large fitness gain from interacting with the other population) are expected to spatially mix (*Momeni et al., 2013*). In contrast, $R^{\leftarrow L}_{\to A}$ and $C^{\leftarrow L}$, a competing pair, should form segregated columns, with each column consisting primarily of a single cell type (*Momeni et al., 2013*). Finally, for the commensal pair $G^{\leftarrow A}_{\to L}$ and $C^{\leftarrow L}$ in which each $G^{\leftarrow A}_{\to L}$ cell can support the local growth of multiple $C^{\leftarrow L}$ cells, $C^{\leftarrow L}$ is expected to grow over $G^{\leftarrow A}_{\to L}$ (*Momeni et al., 2013*). However, it is unclear what patterns would form when the three strains grow together in a community and how the patterns might impact cooperators and cheaters differently.

To examine how cooperation and cheating might affect community patterns, we compared communities grown in a spatial environment under conditions of competition ('Comp') or cooperation and cheating ('Co&Ch') (*Figure 1B*). Specifically, equal proportions of $R^{\leftarrow L}_{\to A}$, $G^{\leftarrow A}_{\to L}$, and $C^{\leftarrow L}$ cells were randomly distributed on top of an agarose surface and allowed to grow. Top-views of experimental ('Exp') communities grown in the presence of adenine and lysine supplements revealed that $R^{\leftarrow L}_{\to A}$, $G^{\leftarrow A}_{\to L}$, and $C^{\leftarrow L}$, when engaged in pure competition, were evenly distributed in the horizontal xy plane (*Figure 3A*, top panel). This pattern was caused by the initial cells growing into microcolonies which, after running into each other, were forced to grow upward (*Figure 3—figure supplement 1A*) (*Momeni et al., 2013*). Indeed, vertical cross-sections exhibited patterns consistent with our expectations that competitive populations should form segregated columns (*Momeni et al., 2013*) (*Figure 3A*, bottom panel).

In the absence of adenine and lysine supplements, the three populations engaged in heterotypic cooperation and cheating in addition to competition for shared resources. Top-views of these experimental communities showed patterns distinct from those of purely competitive communities. Regions dominated by a mixture of the cooperating pair $R^{\leftarrow L}_{\to A}$ and $G^{\leftarrow A}_{\to L}$ appeared isolated from regions dominated

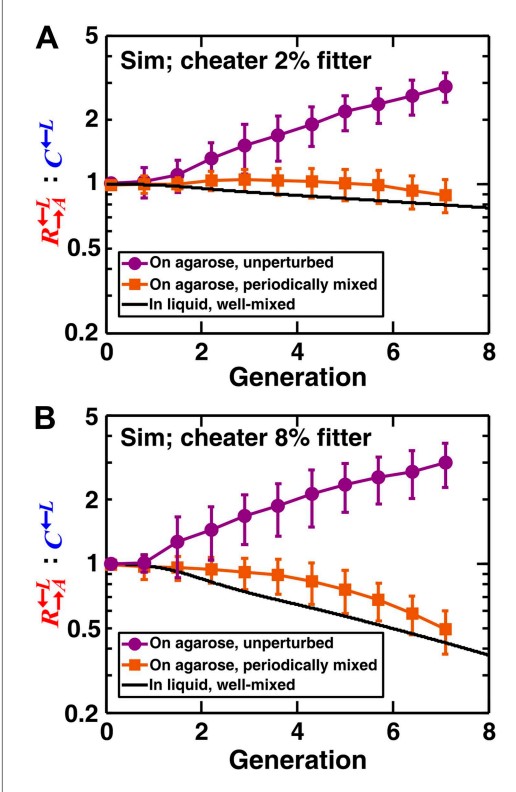

**Figure 2**. In simulated communities, a spatial environment is required to promote heterotypic cooperation. The $R_{\to A}^{\leftarrow L}:C^{\leftarrow L}$ ratios of simulated ('Sim') spatial cooperating and cheating communities were grown either unperturbed (purple circles) or periodically mixed (orange squares). To simulate periodic mixing, the arrangement of cells was completely randomized every 12 hr. In each mixing event, the concentration of adenine and lysine throughout the community was assigned to be the average value over the entire community. Error bars show the standard deviation of ratios in six independent communities. The solid black line shows the ratio in a simulated well-mixed liquid coculture using the same parameters as simulated communities on agarose (***Figure 2—source data 1***). The fitness advantage of cheater over cooperator was either 2% (top panel) or 8% (bottom panel).

The following source data and figure supplements are available for figure 2:

**Source data 1**. Parameter values used in the diffusion model simulations.

**Figure supplement 1**. Stochastic cheater outcomes in periodically mixed communities of ancestral cooperators, cheaters, and partners.

**Figure supplement 2**. A spatial environment is required to favor heterotypic cooperation over cheating.

by the cheater $C^{\leftarrow L}$ (***Figure 3B***, top panel and ***Figure 3—figure supplement 1B***). Vertical cross-sections of these communities revealed that cooperating $R_{\to A}^{\leftarrow L}$ and $G_{\to L}^{\leftarrow A}$ intermixed and formed tall 'pods'. In contrast, cheating $C^{\leftarrow L}$ was relegated to the periphery of these pods and grew relatively poorly (***Figure 3B***, bottom panel).

To quantify differential partner association, the result of partner fidelity feedback, we define 'partner association index' $A_{RG/CG}$: for those $R_{\to A}^{\leftarrow L}$ and $C^{\leftarrow L}$ cells bordering at least one other cell type ('Materials and methods'), $A_{RG/CG}$ is the ratio of the average number of $G_{\to L}^{\leftarrow A}$ in the immediate neighborhood of $R_{\to A}^{\leftarrow L}$ to the average number of $G_{\to L}^{\leftarrow A}$ in the immediate neighborhood of $C^{\leftarrow L}$. A partner association index $A_{RG/CG} > 1$ indicates more $G_{\to L}^{\leftarrow A}$ neighbors surrounding $R_{\to A}^{\leftarrow L}$ than surrounding $C^{\leftarrow L}$. Using a two-dimensional neighborhood to quantify $A_{RG/CG}$ in top-views and vertical cross-sections, we found that $A_{RG/CG}$ significantly exceeded 1 during cooperation and cheating but not during competition (***Figure 3C***). This self-organization—the formation of non-random patterns from initially randomly distributed individuals purely driven by internal local interactions (***Camazine et al., 2003***; ***Solé and Bascompte, 2006***)—automatically makes $G_{\to L}^{\leftarrow A}$ partner more accessible to cooperating $R_{\to A}^{\leftarrow L}$ than to cheating $C^{\leftarrow L}$.

Similar to the experiments, top-views and vertical cross-sections in the communities simulated through the diffusion model ('Sim') also showed that cooperating $R_{\to A}^{\leftarrow L}$ and $G_{\to L}^{\leftarrow A}$ preferentially associated with each other and formed tall pods, while cheating $C^{\leftarrow L}$ was isolated (***Figure 3E***). Such self-organization was absent in pure competition (***Figure 3D***). We quantified the partner association index of simulated communities using a three-dimensional neighborhood averaged across the entire community ($A_{RG/CG}^{3D}$), which turned out to be much less variable than analyzing two-dimensional slices (***Figure 3—figure supplement 2***). During cooperation and cheating, $A_{RG/CG}^{3D}$ increased from an initial value of 1 to a steady level greater than 1 (***Figure 3F***, top). This greater-than-1 $A_{RG/CG}^{3D}$ favored cooperating $R_{\to A}^{\leftarrow L}$ over cheating $C^{\leftarrow L}$, as the ratio $R_{\to A}^{\leftarrow L}:C^{\leftarrow L}$ continued to increase even after $A_{RG/CG}^{3D}$ had leveled off (***Figure 3F***, bottom). In contrast, during pure competition, $A_{RG/CG}^{3D}$ was close to 1 and $C^{\leftarrow L}$ was favored over $R_{\to A}^{\leftarrow L}$ (***Figure 3F***).

## Self-organization can in theory discriminate among cooperators of varying quality

In addition to generating information difficult to obtain from experimental communities (such as the detailed time course of $A_{RG/CG}^{3D}$ described above),

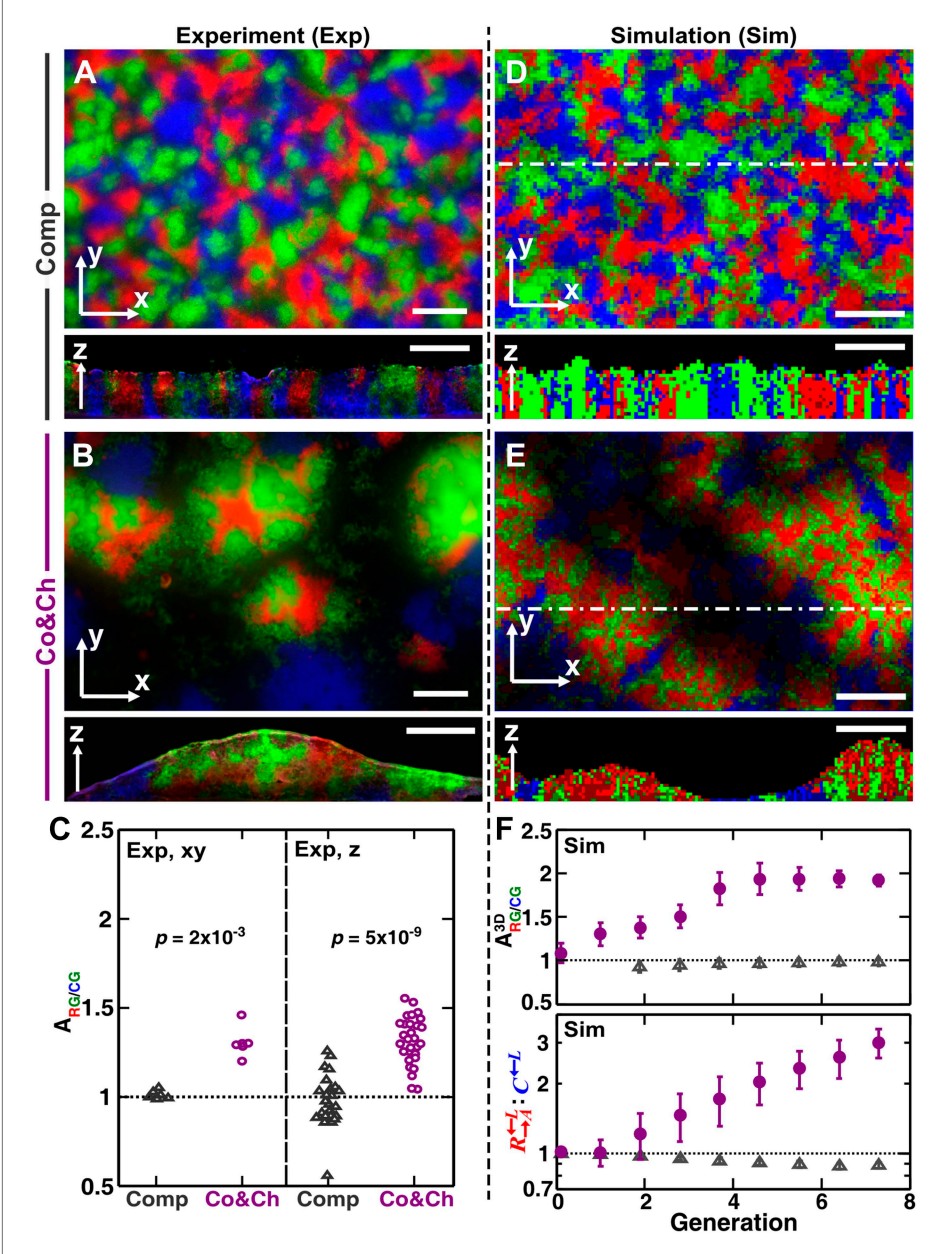

**Figure 3**. Growing cells self-organize to exclude cheaters from heterotypic cooperators. (**A** and **B**) Experimentally, as the initially randomly distributed cells grew, different patterns emerged depending on whether the medium contained or lacked adenine and lysine supplements and consequently whether the dominant cell-cell interaction was respectively competition ('Comp') or cooperation and cheating ('Co&Ch'). Top-views: 'xy'; vertical sections: 'z'. (**C**) Compared to $C^{\leftarrow L}$, $R^{\leftarrow L}_{\rightarrow A}$ had a higher level of association with $G^{\leftarrow A}_{\rightarrow L}$ during cooperation and cheating ($A_{RG/CG} > 1$) but not during competition. In (**C**), the communities were analyzed after approximately six to eight generations. (**D**, **E** and **F**) We observed similar results in the simulated communities. In simulated top-views, higher color intensity indicates a greater number of cells of the corresponding fluorescent color stacked at that position. In simulated vertical cross-sections, low and high color intensity represent dead and live cells, respectively. Scale bar: 100 μm. In **C** and **F**, grey: competition; magenta: cooperation and cheating. p Values are from the Mann–Whitney U-test. All communities started from an initial density of 3000 total cells/mm² (*Figure 1—figure supplement 1A*).

The following figure supplements are available for figure 3:

**Figure supplement 1**. Cooperation-cheating and pure competition led to distinct community patterns in top-views.

*Figure 3. Continued on next page*

*Figure 3. Continued*

**Figure supplement 2**. The partner association indexes in simulated communities showed more association between $R^{←L}_{→A}$ and $G^{←A}_{→L}$ than between $C^{←L}$ and $G^{←A}_{→L}$ during cooperation and cheating but not during competition.

simulations enabled us to explore a broader class of cooperator–cheater communities. Simulations showed that self-organization also allowed discrimination among cooperators of varying quality (**Figure 4**). Specifically, we initiated diffusion-model simulations using three populations: $G^{←A}_{→L}$, $R^{←L}_{→A}$, and $R^{←L}_{→A,\textbf{d}}$. The adenine release rate of $R^{←L}_{→A,\textbf{d}}$ was a fraction **d** (< 1) of that of $R^{←L}_{→A}$, and like $C^{←L}$, $R^{←L}_{→A,\textbf{d}}$ had a constant growth rate advantage over $R^{←L}_{→A}$ at all lysine concentrations. When grown in the absence of

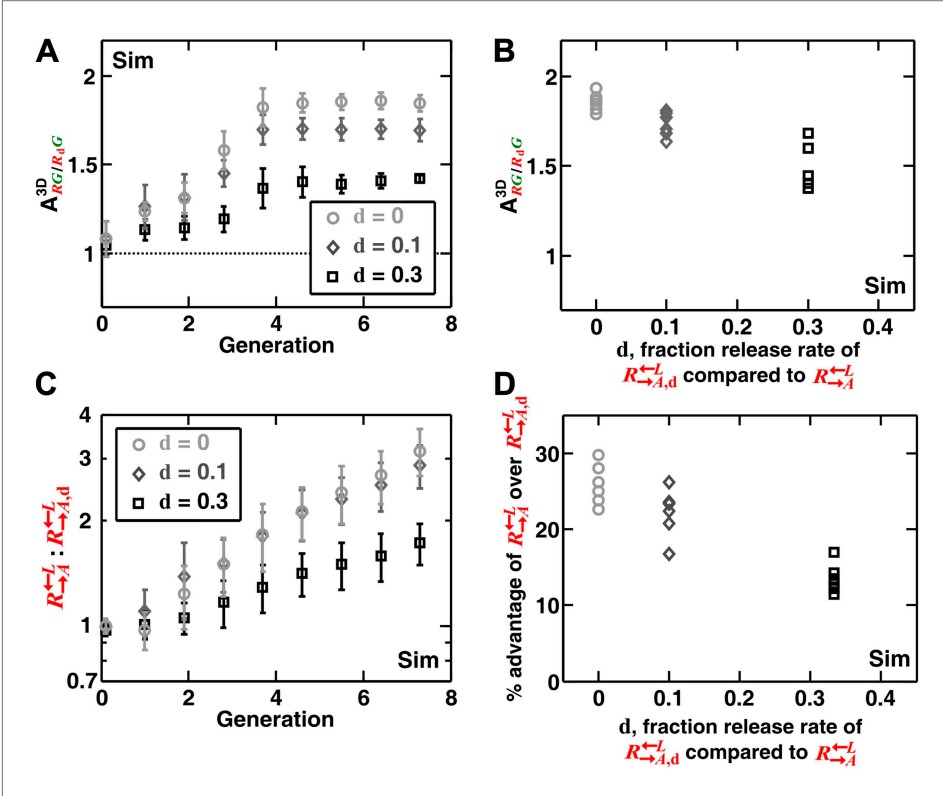

**Figure 4**. Self-organization leads the most giving cooperator to associate most with the heterotypic cooperative partner, allowing discrimination of cooperators of varying quality. In diffusion model simulations, $R^{←L}_{→A,\textbf{d}}$ produced adenine at a rate **d**-fold (0 ≤ **d** < 1) of the release rate of $R^{←L}_{→A}$. $R^{←L}_{→A,\textbf{d}}$ had a 5% fitness advantage over $R^{←L}_{→A}$ at all lysine concentrations. $R^{←L}_{→A,\textbf{d}}$ that released less (smaller values of **d**) were isolated more in spatial patterns (**A**, steady-state values summarized in **B**) and disfavored more as the community grew (**C** and **D**). In (**D**), the fitness advantage of $R^{←L}_{→A}$ over $R^{←L}_{→A,\textbf{d}}$ was calculated from the rate of changes in the ratio $R^{←L}_{→A}$:$R^{←L}_{→A,\textbf{d}}$ between generations 2 and 6 in (**C**). In (**B**) and (**D**), data from six replicates were plotted. The communities were initiated at 3000 total cells/mm². Sim: simulation.

The following figure supplements are available for figure 4:

**Figure supplement 1**. Spatial self-organization allowed $R^{←L}_{→A}$ to increase in frequency even when $R^{←L}_{→A,\textbf{d}=0}$ was much fitter.

**Figure supplement 2**. Cheaters with a much higher affinity for cooperative benefits than cooperators can destroy heterotypic cooperation even in a spatial environment.

**Figure supplement 3**. In obligatory byproduct mutualism (benefit production incurs no cost and non-producers have no fitness advantage over producers), high levels of non-producers can still destroy byproduct mutualism in a spatial environment.

supplements, $R^{\leftarrow L}_{\to A,\mathbf{d}}$ were isolated and outcompeted by $R^{\leftarrow L}_{\to A}$. This effect was quantitative, as a lower $\mathbf{d}$ value resulted in a larger $A_{RG/R_dG}$ (i.e., the less $R^{\leftarrow L}_{\to A,\mathbf{d}}$ released, the more it was excluded; *Figure 4A,B*), and a greater advantage of $R^{\leftarrow L}_{\to A}$ over $R^{\leftarrow L}_{\to A,\mathbf{d}}$ (*Figure 4C,D*). In contrast, $A_{RG/R_dG}$ was not highly sensitive to the intrinsic growth rate advantage of $R^{\leftarrow L}_{\to A,\mathbf{d}}$ over $R^{\leftarrow L}_{\to A}$ (*Figure 4—figure supplement 1*). $R^{\leftarrow L}_{\to A,\mathbf{d}}$ was still strongly disfavored even when it had relatively large (50%) intrinsic growth rate advantage over $R^{\leftarrow L}_{\to A}$ (*Figure 4—figure supplement 1*). Taken together, self-organization favors cooperators that supply the most benefits.

## Heterogeneity in the initial spatial distribution of cells facilitates but is not required for self-organization

One might hypothesize that among the initially randomly distributed cell populations, the occasional fortuitous proximity of a cooperator cell to a partner cell is necessary for self-organization. To examine this hypothesis, we simulated communities starting from a periodic and symmetric initial distribution of cells (*Figure 5A*). In the absence of initial spatial asymmetry, other stochastic effects such as the initial metabolite-storage state of the cell, cell rearrangement, and death events broke the symmetry and self-organization still emerged (*Figure 5B*). Even though the final partner association index from random

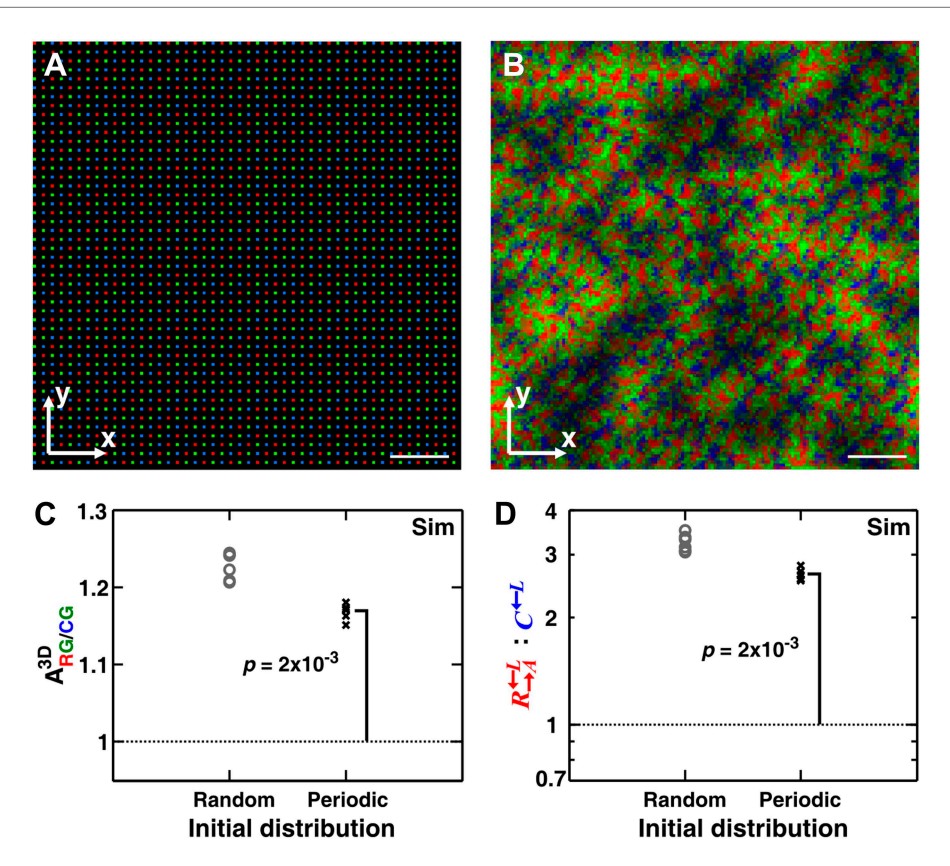

**Figure 5**. Heterogeneity in the initial spatial distribution of cells facilitates but is not required for cheater isolation. Starting from a symmetric and periodic distribution in which all cooperators $R^{\leftarrow L}_{\to A}$ and cheaters $C^{\leftarrow L}$ had an equal access to the partner $G^{\leftarrow A}_{\to L}$ (**A**), heterotypic cooperators self-organized (**B** and **C**) and were favored (**D**). (**B**–**D**) corresponded to generation 6. Break of symmetry from the initial symmetric spatial distribution can be due to stochastic effects such as differences in the initial amounts of metabolites cells possessed, death of cells, or the random direction of cell division. Compared to a random initial distribution, communities with a periodic initial distribution showed smaller mean $A_{RG/CG}$; nonetheless, $A_{RG/CG}$ significantly exceeded 1 (Wilcoxon signed rank test). In these simulations, the growth rate advantage of $C^{\leftarrow L}$ over $R^{\leftarrow L}_{\to A}$ was assumed to be 10% at all concentrations of lysine. The communities were initiated at 4400 total cells/mm². Scale bar: 100 µm. In simulated top-views, higher color intensity indicates a greater number of cells of the corresponding color stacked at that position. Sim: simulation.

initial distribution was greater than that from periodic distribution, the latter was still significantly greater than 1 (*Figure 5C*). Consequently, cooperators still outperformed cheaters in periodic initial distribution, even though the degree of outperformance was greater in random initial distribution (*Figure 5D*). These simulation results suggest that the heterogeneity in the initial spatial distribution of cells promoted but was not required for self-organization.

## Asymmetric fitness effects of cooperators and cheaters on partners drives self-organization

What drives differential partner association during self-organization? Since self-organization was only observed during cooperation and cheating but not during competition, the very acts of cooperation and cheating are required. When the fitness effects of interactions are the major driving force of patterning, interacting populations are expected to intermix if both supply spatially localized large fitness benefit to the other (*Momeni et al., 2013*). In contrast, lack of benefit to either population causes population segregation (*Momeni et al., 2013*). Thus, we reason that the asymmetry between cooperators and cheaters in their capacity to reciprocate partner's benefits drives self-organization.

To examine how this asymmetry leads to self-organization, we simulated a community in which a center stripe of partners was initially bordered by a stripe of cooperators on one side and cheaters on the other (*Figure 6*). This simulation configuration allowed us to examine the process of self-organization from the simplest form of initial symmetry. Near the cooperator side, partners intermixed with cooperators due to the spatially localized large benefits to both (*Momeni et al., 2013*; red and green in *Figure 6*). In contrast, near the cheater side, the lack of benefits to partners caused minimal intermixing between cheaters and partners (*Momeni et al., 2013*; blue and green in *Figure 6*). This isolation of cheaters allowed cooperators to increase in frequency despite the intrinsic fitness advantage of cheaters over cooperators.

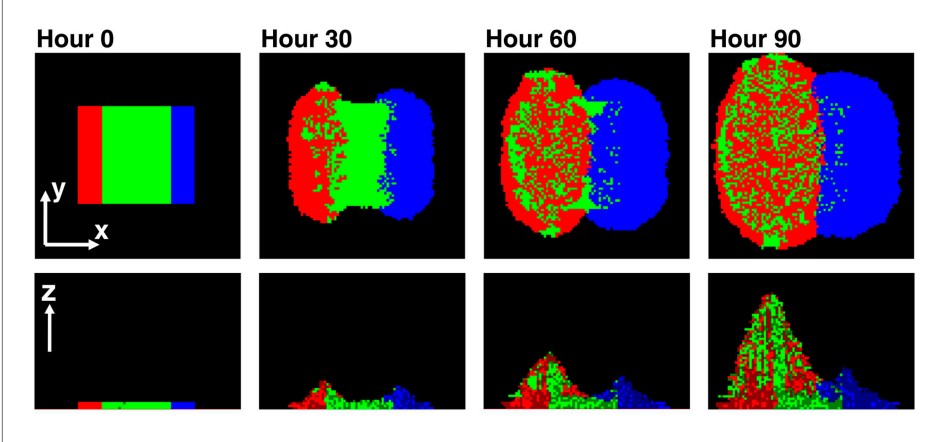

**Figure 6**. Asymmetric fitness effects of cooperators and cheaters on partners drive self-organization. Time progression of self-organization in a simulated community as observed in top-views (top) and vertical cross-sections (bottom). Heterotypic cooperative partners (green) supply large benefits to both cooperators (red) and cheaters (blue). Since the benefit is spatially localized, only cooperators and cheaters that are close to partners will grow. Given that cells dividing toward partners will on average have more access to benefits than those dividing away from partners, both cooperators and cheaters pile over partners. Cooperators reciprocate by supplying a large, but different, localized benefit to the partner, while cheaters do not. Thus, partners grow and pile over cooperators but not cheaters. Consequently, further growth of cooperators is facilitated, while cheaters become isolated and disfavored. In this simulation, cheaters have an 8% fitness advantage over cooperators. To mimic the top-view from microscopy, the top-view in this simulation represents the top-most layer of cells instead of the total intensity integrated over z at each pixel.

The following figure supplements are available for figure 6:

**Figure supplement 1**. Localization of cooperative benefits is required for self-organization.

## Cell growth into open space is required for self-organization

If the cell densities in the initial inoculum were so high that all cell types were forced to be each other's immediate neighbor, then the degree of self-organization might be limited. To test whether access to open space is required for self-organization, we initiated communities at high initial cell densities such that two cell layers covered the inoculation spot. We then allowed communities to grow unperturbed in a spatial environment (*Figure 1—figure supplement 1B*). Compared to the center, the expanding front where open space offered opportunities for self-organization showed a significantly higher partner association index (*Figure 7*, *Figure 7—figure supplement 1*). In the community center, cooperators were not favored over cheaters. In contrast, $R_{\rightarrow A}^{\leftarrow L}:C^{\leftarrow L}$ increased progressively above the initial value of 1 as the community grew into open space away from the inoculum (*Figure 7*). Thus, similar to what has been observed for homotypic cooperation (*Datta et al., 2013*; *Van Dyken et al., 2013*), growth into open space during range expansion also favors heterotypic cooperation over cheating.

## Discussion

Using an engineered yeast community and a mathematical model devoid of partner recognition, we examined how partner-fidelity feedback unfolds in a spatial environment at the individual cell level. In our system, partner cells released essential metabolites for cooperators and cheaters, and cooperators reciprocated with a different essential metabolite while cheaters did not. We found that despite an initially random or periodic spatial distribution, cells 'self-organized' into a non-random and non-symmetric pattern: cooperators had more partner neighbors than cheaters did. The level of differential partner association, as quantified by the partner association index, is correlated with how much cooperators outperform cheaters despite the intrinsic fitness advantage of cheaters over cooperators.

What is required for self-organization? Self-organization is driven by the asymmetry between cooperators and cheaters in the amount of spatially localized benefits they supply to the heterotypic partner during cell growth into open space. Our previous work has shown that if two distinct populations receive spatially localized large fitness benefits from each other, then the two populations are expected to intermix (*Momeni et al., 2013*). In contrast, when the fitness benefit to at least one population is small, then cell types are not expected to intermix (*Momeni et al., 2013*). Consequently, partners intermix with cooperators but not cheaters (*Figures 3–7*). This difference in the tendency to intermix causes differential partner association, which facilitates the growth of cooperators and isolates and disfavors cheaters. Indeed in simulations, if intermixing was prevented through delocalizing benefits (*Momeni et al., 2013*; *Allen et al., 2013*), or if intermixing was imposed on all cells through high initial cell densities, self-organization and cooperator advantage over cheaters diminished (*Figure 6—figure supplement 1*, *Figure 7*). Similar trends are observed in recent works which mathematically examined diffusion of public good in homotypic cooperation (*Allen et al., 2013*; *Borenstein et al., 2013*).

Simulations showed that spatial self-organization could discriminate among cooperators that supply different levels of benefits (*Figure 4*). When cooperative benefits were limiting, the most 'helpful' cooperator ended up with the highest number of partners and was most favored. In our mathematical model, we could be sure that this was not due to partner recognition. Thus, in theory, self-organization through partner fidelity feedback is capable of achieving finer levels of discrimination without requiring sophisticated cognition and memory. Without knowing the molecular mechanisms of interactions, one could easily confuse partner fidelity feedback with partner recognition, because both are capable of discriminating partners of varying cooperative qualities. Our work suggests that any argument on partner recognition in a system that is intrinsically spatial (such as legume and rhizobia and fig and colonizing fig wasp) will require identifying variants that fail to distinguish cooperators from cheaters. Otherwise, interpreting cheater discrimination as 'partner choice' can be misleading.

Spatial self-organization can fend off cheaters with large fitness advantages over cooperators (*Figure 4D*). However, if (non-isogenic) cheaters are much better than cooperators at taking up low concentrations of benefits, then cheaters will destroy heterotypic cooperation even in a spatial environment (experimental results in *Figure 4—figure supplement 2*). In byproduct mutualism (benefit production incurs no cost and thus non-producers have no fitness advantage over producers), non-producers

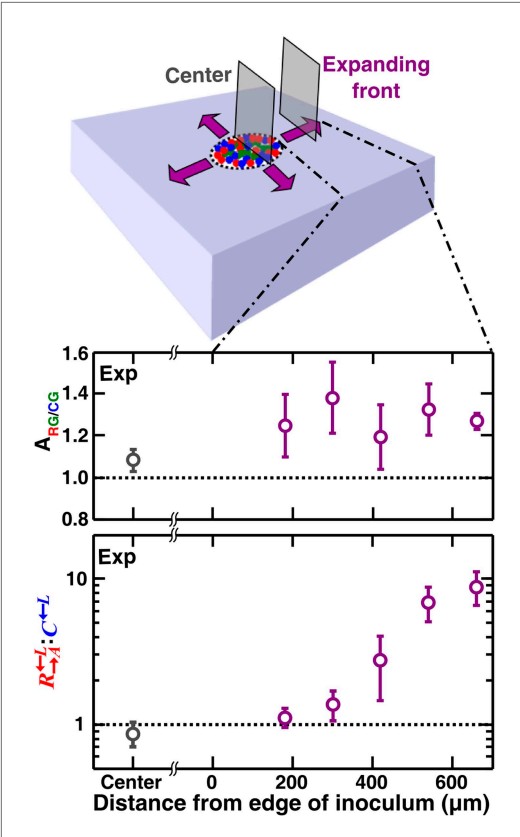

**Figure 7**. Cell growth into open space is required for self-organization and cheater isolation. In an unperturbed spatial environment, a community starting from a high-density (~$10^5$ total cells/mm$^2$) confluent inoculum expanded to new territories (purple). Compared to 'Center', self-organization was significantly more in 'Expanding front'. Consequently, the initially 1:1 $R_{\rightarrow A}^{\leftarrow L}{:}C^{\leftarrow L}$ ratio changed in favor of cooperators as the community expanded outward but not in 'Center'. Sections were collected from three independent communities. Using the Mann–Whitney $U$-test, the association indexes of the 'Center' and of the 'Expanding front' were significantly different (p = 4 × 10$^{-4}$). Exp: experiment.

The following figure supplements are available for figure 7:

**Figure supplement 1**. Vertical cross-sections of communities started from a high-density inoculum.

are excluded and disfavored in a spatial environment (*Figure 4—figure supplement 3*). Furthermore, high levels of non-producers can still destroy byproduct mutualism in a spatial environment (simulation results in *Figure 4—figure supplement 3*). This is because non-producers compete with producers for benefits from the partner. If the initial level of non-producers is too high, producers receive few benefits and suffer, which in turn negatively impacts the partner.

Conceptually, spatial self-organization favoring heterotypic cooperation can be considered as an example of niche construction or environmental feedback (*Lehmann, 2007*; *Pepper, 2007*). Cooperators construct favorable niches and cheaters construct unfavorable niches for partners. Through differential partner association, partners construct better niches for cooperators than for cheaters. Niche construction not only affects the growth of current cells, but also that of the future progeny. This reciprocal niche construction favors heterotypic cooperation over cheating.

In summary, spatial self-organization is the mechanism for partner fidelity feedback. Not requiring the evolution of partner recognition (*Travisano and Velicer, 2004*; *Foster and Wenseleers, 2006*), spatial self-organization offers a simple and fundamental mechanism solely driven by acts of strong cooperation and cheating during cell growth into open space. Even in natural heterotypic cooperative systems with long evolutionary histories, self-organization may act either alone or in synergy with potential recognition mechanisms (*Weyl et al., 2010*) to exclude cheaters.

## Materials and methods

### Engineered yeast strains

$R_{\rightarrow A}^{\leftarrow L}$, $G_{\rightarrow L}^{\leftarrow A}$, and $C^{\leftarrow L}$ were respectively WS950 (*MATa ste3::kanMX4 lys2Δ0 ade4::ADE4(PUR6) ADHp-DsRed.T4*), WS954 (*MATa ste3::kanMX4 ade8Δ0 lys21::LYS21(fbr) ADHp-venus-YFP*), and WS962 (*MATα ste3::kanMX4 lys2Δ0 ADHp-CFP*).

### Community growth and measurements

To grow yeast communities, agarose columns were prepared by pouring 2× concentrated SD minimal medium (*Guthrie and Fink, 1991*) with 2% low melting temperature agarose either into flat-bottom 96-well plates (*Figure 1—figure supplement 1A*) or into rectangular petri dishes (*Figure 1—figure supplement 1B*). Agarose in the rectangular petri dishes was subsequently cut into 24 mm × 24 mm × 4 mm pads. For competition experiments, 2× SD was supplemented with lysine and adenine (650 μM and 430 μM final concentrations, respectively). The communities on agarose were inoculated either from a uniform distribution of all cells (*Figure 1—figure supplement 1A* at 3000 or 10,000 total cells/mm$^2$) or a high-density inoculum of ~2 mm diameter (*Figure 1—figure supplement 1B* at ~8 × 10$^4$ total cells/mm$^2$), as specified.

We grew well-mixed liquid cocultures in 3 ml of 1× SD without supplementing adenine or lysine, with initially $5 \times 10^5$ total cells/ml. In all cases, cultures were initiated from equal proportions of $R_{\to A}^{\leftarrow L}$, $G_{\to L}^{\leftarrow A}$, and $C^{\leftarrow L}$ populations. Flow cytometry was used to measure population sizes of different types in a community (*Momeni et al., 2013*). Fluorescent imaging equipment and procedures are described in *Momeni et al. (2013)*. Cryosectioning to obtain vertical cross-sections of yeast communities followed *Momeni and Shou (2012)*.

## Quantification of partner association

We quantified the relative association of cooperators and cheaters with the heterotypic partner by dividing the average number of immediate $G_{\to L}^{\leftarrow A}$ neighbors per focal $R_{\to A}^{\leftarrow L}$ cell by the average number of immediate $G_{\to L}^{\leftarrow A}$ neighbors per focal $C^{\leftarrow L}$ cell when a focal cell neighbored at least one different population ($A_{RG/CG}$, partner association index). We chose to count the immediate $G_{\to L}^{\leftarrow A}$ neighbors because nearest neighbors presumably had the greatest impact on the growth of the focal cell due to spatially localized benefits. We chose to analyze focal cells neighboring at least one different population because cells surrounded by their own types do not contribute as much to population growth as cells surrounded by partners.

In simulations, a three-dimensional neighborhood around each cell was used to quantify the association index ($A_{RG/CG}^{3D}$), whereas in experiments, a two-dimensional neighborhood was used in two-dimensional top-views or cross-sections of the community. Based on fluorescence intensities in the DsRed, YFP, and CFP channels, cell types were assigned to each pixel in fluorescent images. Pixels having fluorescence intensities less than 30% above the background in all fluorescence channels were defined as 'no signal'. Otherwise, fluorescence intensities in each channel were normalized to their respective image-wide 90th percentile values and pixel identity was assigned to be the same as the fluorescence channel with the highest normalized intensity. For cross-sections taken from the center of communities in *Figure 7*, the top crown of cross-sections appeared very bright, whereas the middle and lower regions appeared dim (*Figure 7—figure supplement 1*). This large dynamic range caused artifacts in cell identification. The intensity thresholds in these sections were therefore manually adjusted to increase the accuracy of cell identification. Eliminating manual adjustments did not alter the conclusions.

## The diffusion model

To simulate the growth of three-dimensional yeast communities, we used the agent-based diffusion model (*Momeni et al., 2013*). In this model, metabolites are released by cooperators and partners, diffuse throughout the community and agarose, and are consumed by cells that need the metabolite. Most parameters were measured experimentally (*Figure 2—source data 1*). We provide a summary of the most relevant features of the model below, without repeating the implementation details that can be found in *Momeni et al. (2013)*.

Cells take up their required metabolites depending on the local concentration of the metabolite according to the Michaelis–Menten equation:

$$v_i(S_i) = v_{m,i} \frac{S_i}{S_i + K_i},$$

where, $i = R_{\to A}^{\leftarrow L}$, $G_{\to L}^{\leftarrow A}$, or $C^{\leftarrow L}$ corresponding to each cell type, $S_i$ is the concentration of the required metabolite for each cell type ($S_i = S_L$, lysine concentration for $R_{\to A}^{\leftarrow L}$ and $C^{\leftarrow L}$, and $S_i = S_A$, adenine concentration for $G_{\to L}^{\leftarrow A}$), $v_{m,i}$ is the maximum uptake rate when metabolites are abundant, and $K_i$ is assumed to be equal to the Monod constant (the concentration of limiting nutrient at which half maximal growth rate is achieved) for each cell type. $R_{\to A}^{\leftarrow L}$ and $C^{\leftarrow L}$ cells require $\alpha_L$ fmole of lysine and $G_{\to L}^{\leftarrow A}$ cells require $\alpha_A$ fmole of adenine to produce a new daughter cell.

The distribution of metabolites is modeled using the diffusion equation, with uptake and release as sinks and sources. Using simplified notations of $S_A = S_A(t)$ and $S_L = S_L(t)$ at time $t$, the metabolite distributions after a time step $t_u$ are calculated as

$$S_A(t+t_u) = S_A + t_u[\nabla \cdot (D\nabla S_A) - v_{m,G} \frac{S_A}{S_A + K_A} n_G + \gamma_A n_R], \text{ and}$$

$$S_L(t+t_u) = S_L + t_u\left[\nabla \cdot (D\nabla S_L) - v_{m,R} \frac{S_L}{S_L + K_{MM,R}} n_R - v_{m,C} \frac{S_L}{S_L + K_{MM,C}} n_C + \frac{\beta_L}{t_u} n_G'\right].$$

Here, $n_R$, $n_C$, and $n_G$ are, respectively, the densities of live $R^{\leftarrow L}_{\rightarrow A}$, $C^{\leftarrow L}$, and $G^{\leftarrow A}_{\rightarrow L}$ cells in a focal community grid. $\nabla = \frac{\partial}{\partial x}\hat{x} + \frac{\partial}{\partial y}\hat{y} + \frac{\partial}{\partial z}\hat{z}$ is the vector differential operator. $n'_G$ is the density of $G^{\leftarrow A}_{\rightarrow L}$ cells that die (which is distributed as binomial ($n_G$, $p$) where $p = t_u \cdot d_G$, with $d_G$ being the death rate of $G^{\leftarrow A}_{\rightarrow L}$) and release lysine in the time step $t_u$. $D$ is a spatially varying function representing the diffusion coefficient in the environment (the value of $D$ was 360 μm²/s inside agarose, 20 μm²/s inside yeast communities, and 0 μm²/s in the surrounding air, according to experimental measurements). Considering $D$ as a spatially varying function simplifies the numerical calculations by automatically incorporating the boundary conditions at the community–air interface. $v_{m,i}$, the maximum uptake rate of the limiting nutrient per cell, relates to the maximum growth rate $r_{m,i}$ through $v_{m,i} = \alpha_i r_{m,i}/\ln 2$, where $\alpha_i$ is the amount of limiting nutrient required to produce a new cell ($\alpha_R = \alpha_C = \alpha_L$ and $\alpha_G = \alpha_A$). $C^{\leftarrow L}$ has a fixed intrinsic fitness advantage over $R^{\leftarrow L}_{\rightarrow A}$ at all lysine concentrations. This advantage was modeled as a higher uptake rate for the limiting nutrient. For example, a 2% fitness advantage of cheaters means $v_{m,C} = 1.02\,v_{m,R}$ or equivalently $r_{m,C} = 1.02\,r_{m,R}$. $\beta_L$ is the amount of lysine released upon the death of a $G^{\leftarrow A}_{\rightarrow L}$ cell, and $\gamma_A$ is the release rate of adenine per $R^{\leftarrow L}_{\rightarrow A}$ cell. $C^{\leftarrow L}$ cheaters do not release any adenine. For other types of cheaters that release adenine with a lower rate compared to cooperators (e.g., in *Figure 4*), a corresponding release term is included in the equation.

To solve this diffusion equation, we follow the above finite difference time-domain equations over time. The diffusion equation is solved over two separate spatial domains (*Momeni et al., 2013*), one containing the agarose (with a 60 μm grid size), and the other containing the community and the air above it (with a 15 μm grid size). These grid sizes accommodated different diffusion coefficients in agarose and in community and represented the average distance nutrient molecules diffuse in 3.5 s. When we used the same diffusion coefficient (360 μm²/s and one grid size of 50 μm) for agarose and community, similar results were obtained. No-flow ($\partial S_i/\partial z = 0$) boundary conditions are applied to the top and bottom surfaces of the simulation domain and periodic boundary conditions are applied to the four vertical sides of the domain.

To incorporate the effects of competition for other shared resources, the growth of all cells also depended on, in addition to adenine or lysine, a shared resource (for instance, glucose) that was initially supplied in the medium (*Figure 2—source data 1*). In such simulations, diffusion and uptake of glucose were also simulated in a way similar to the above equations. Each cell divided only after acquiring enough glucose, in addition to adenine or lysine. Once a cell had accumulated one metabolite sufficient for one cell division, it stopped consuming that metabolite and continued to acquire the second metabolite until a sufficient amount had been acquired to trigger the birth of a daughter cell.

In simulations, $R^{\leftarrow L}_{\rightarrow A}$, $C^{\leftarrow L}$, and $G^{\leftarrow A}_{\rightarrow L}$ cells are initially randomly distributed on the surface of solid medium. The cells start from random initial storage of their required metabolites. In each $t_u$ time step, each live cell takes up its required metabolites according to the Michaelis–Menten equation shown above. Each cell type is assumed to require its limiting metabolite and a shared metabolite ($\alpha_L$ lysine for $R^{\leftarrow L}_{\rightarrow A}$ and $C^{\leftarrow L}$, $\alpha_A$ adenine for $G^{\leftarrow A}_{\rightarrow L}$, and $\alpha_G$ for the shared glucose for all cell types, all listed in *Figure 2—source data 1*) to divide. The state of cells is examined at every $\tau$ time interval ($\tau = 6$ min, which contains several diffusion $t_u$ time steps, but is still much shorter than the minimum cell doubling time of ~2 hr). The cells that have acquired the required amount of limiting metabolites divide.

Cell divisions in a three-dimensional community often required cell rearrangement. Assumptions concerning cell rearrangement were derived from experimental observations (*Momeni et al., 2013*). Time-lapse images of the growth of a single fluorescent cell into a microcolony showed that the center of the microcolony became brighter due to multiple cell layers when the microcolony grew to larger than a five-cell radius (*Momeni et al., 2013*). Thus, we assumed that each cell initially budded in the horizontal plane and pushed others in its immediate neighborhood to the side along the shortest path to empty space. Once a cell was completely surrounded on each side by roughly five cells, it either budded directly upward with a probability of 70% or randomly budded to one of the sides at the same level, pushing up the displaced cell and all the cells above. The probability of 0.7 (instead of 1) of dividing directly upward is estimated from experimental observations: when individual green-fluorescent cells were surrounded by many equally fit competing red-fluorescent cells, vertical cross-sections showed that as z increased, the progeny of the green-fluorescent cell 'diffused'

laterally (instead of remaining a vertical line) (*Figure 3—figure supplement 1F* in *Momeni et al., 2013*). Since we cannot experimentally track all possible outcomes of cell rearrangement, this assumption is a simplification of reality. This simplification could have contributed to the discrepancy between experimental and simulation patterns, although our conclusions from experiments and simulations are similar.

The cells also die stochastically corresponding to their fixed death rates ($d_R$, $d_G$, or $d_C$ as listed in *Figure 2—source data 1*). After each cell state update ($\tau$), the diffusion coefficient is updated: the diffusion coefficient in each community diffusion grid (15 µm × 15 µm × 15 µm, maximally containing 3 × 3 × 3 = 27 cells of size 5 µm × 5 µm × 5 µm) is assumed to be proportional to the occupancy of that grid, changing from 0 to 20 µm²/s. In this three-dimensional agent-based model of community growth, the initial conditions of cells, cell death, random direction of growth, and cell rearrangement are the only sources of stochasticity; metabolite uptake and diffusion of metabolites in the environment are modeled as deterministic phenomena.

Most of the parameters used for the simulations (*Figure 2—source data 1*) are measured experimentally. More details of the implementation and assumptions can be found in *Momeni et al. (2013)*. An example of the implementation of this model as a MATLAB code is included in additional files (*Source code 1*).

## Yeast strains adapted to low-nutrient conditions

When the ancestral $R^{\leftarrow L}_{\rightarrow A}$, $G^{\leftarrow A}_{\rightarrow L}$, and $C^{\leftarrow L}$ were periodically mixed on an agarose pad lacking adenine and lysine, the final $R^{\leftarrow L}_{\rightarrow A}$:$C^{\leftarrow L}$ ratios were stochastic, either in favor of cheaters or cooperators (*Figure 2—figure supplement 1*). This is consistent with previous experiments in liquid cocultures (*Waite and Shou, 2012*) that also yielded stochastic cheater outcomes due to an adaptive race between $R^{\leftarrow L}_{\rightarrow A}$ and $C^{\leftarrow L}$. During the adaptive race, both $R^{\leftarrow L}_{\rightarrow A}$ and $C^{\leftarrow L}$ sampled from the same set of mutations that enhanced cell fitness in the lysine-limited cooperative environment. The population with the fittest mutant rapidly dominated the coculture (*Figure 2—figure supplement 1*; *Waite and Shou, 2012*).

To mitigate the confounding effect of adaptive race, we used $R^{\leftarrow L}_{\rightarrow A}$ and $C^{\leftarrow L}$ populations preadapted to the cooperative environment. First, $R^{\leftarrow L}_{\rightarrow A}$ containing a mutation in *RSP5* (CT8, see Table 1 in *Waite and Shou, 2012*), known to significantly improve the fitness of lysine-requiring cells under lysine-limitation (*Waite and Shou, 2012*), was crossed to the ancestral $C^{\leftarrow L}$ to produce a diploid (WS1421). Sporulation of the diploid yielded a cyan-fluorescent cooperator (WS1447) and a red-fluorescent cheater (WS1448), both harboring the *rsp5* mutation. The temperature sensitivity of this *rsp5* allele allowed its easy selection at 37°C. To avoid confusion, we indicate these cooperators and cheaters as *rsp5* $R^{\leftarrow L}_{\rightarrow A}$ and *rsp5* $C^{\leftarrow L}$, respectively. Several well-mixed cocultures consisting of *rsp5* $R^{\leftarrow L}_{\rightarrow A}$, *rsp5* $C^{\leftarrow L}$, and the ancestral partner $G^{\leftarrow A}_{\rightarrow L}$ (WS954) were initiated at a ratio of 1:1:1. The initial stochastic phase in population dynamics was indicative of additional rounds of adaptive races (*Waite and Shou, 2012*) between *rsp5* $R^{\leftarrow L}_{\rightarrow A}$ and *rsp5* $C^{\leftarrow L}$ (*Figure 2—figure supplement 2A*). After 250 hr, the $R^{\leftarrow L}_{\rightarrow A}$:$C^{\leftarrow L}$ ratios showed steady trends, suggesting the absence of further rapid adaptive races (*Figure 2—figure supplement 2A*). After ~500 hr, two of the lines (brown) that displayed a steady change of the *rsp5* $R^{\leftarrow L}_{\rightarrow A}$:*rsp5* $C^{\leftarrow L}$ ratio in favor of cheaters and a final ratio close to 1:1 were frozen down. After reviving these preadapted cocultures (hereafter marked as " ′ "), $R'^{\leftarrow L}_{\rightarrow A}$:$C'^{\leftarrow L}$ continued to decline steadily (*Figure 2—figure supplement 2B*). Propagating well-mixed liquid cocultures from these lines for an additional 400 hr, we observed that the $R'^{\leftarrow L}_{\rightarrow A}$:$C'^{\leftarrow L}$ ratio exhibited a steady decline (*Figure 2—figure supplement 2C*, 'In liquid, well-mixed'), suggesting a cheater $C'^{\leftarrow L}$ fitness advantage of around 8% (7.5 ± 0.8% SD) over $R'^{\leftarrow L}_{\rightarrow A}$. It should be noted that since $R'^{\leftarrow L}_{\rightarrow A}$ and $C'^{\leftarrow L}$ are not isogenic, the fitness advantage of $C'^{\leftarrow L}$ over $R'^{\leftarrow L}_{\rightarrow A}$ is likely not solely due to the lack of adenine overproduction by $C'^{\leftarrow L}$. Nevertheless, this case models an advantage for the cheating type over the cooperating type, similar to what might be observed in nature if cheaters and cooperators are of different species (*Côté and Cheney, 2005*; *Jandér and Herre, 2010*).

## Acknowledgements

We would like to thank Chi-Chun Chen, David Skelding, Robin Green, Alexander Pozhitkov, Ben Kerr, Harmit Malik, Katie Peichel, Suzannah Rutherford, Kevin Foster, and Jim Bull for their suggestions. We thank Aric Capel for constructing the WS1447 and WS1448 yeast strains. We thank the reviewing editor and anonymous reviewers for their constructive feedback and suggestions.

## Additional information

### Funding

| Funder | Grant reference number | Author |
| --- | --- | --- |
| National Institute of Health | 1 DP2 OD006498-01 | Babak Momeni, Wenying Shou |
| W M Keck Foundation | | Adam James Waite, Wenying Shou |
| Gordon and Betty Moore Foundation | GBMF 2550.01 | Babak Momeni |
| Life Sciences Research Foundation | | Babak Momeni |

The funders had no role in study design, data collection and interpretation, or the decision to submit the work for publication.

### Author contributions

BM, Conception and design, Acquisition of data, Analysis and interpretation of data, Drafting or revising the article; AJW, Conception and design, Acquisition of data, Drafting or revising the article; WS, Conception and design, Analysis and interpretation of data, Drafting or revising the article

## Additional files

### Supplementary files

• Source code 1. A MATLAB code for simulating the progression of patterns in a community of $R_{\to A}^{\leftarrow L}$, $G_{\to L}^{\leftarrow A}$, and $C^{\leftarrow L}$. The code is based on the diffusion model ('Materials and methods') and includes initially supplied glucose in the medium as a shared resource consumed by all populations (parameters in *Figure 2—source data 1*).

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
