## [Decision Letter]

Thank you for sending your work entitled “Spatial Self-organization Favors Heterotypic Cooperation over Cheating” for consideration at *eLife*. Your article has been evaluated by a Senior editor and 3 reviewers, one of whom is a member of our Board of Reviewing Editors.

The Reviewing editor and the other reviewers discussed their comments before we reached this decision, and the Reviewing editor has assembled the following comments to help you prepare a revised submission.

The study is based on a previously developed system of two yeast strains that can enter cooperative interactions. The authors have already published an *eLife* paper (http://dx.doi.org/10.7554/eLife.00230) describing a spatial patterning model in conjunction with a series of experiments that aimed to study inter-population cooperation. The present paper deals with the question of how cooperating patterns could emerge in the presence of a cheater. They use a similar technological approach as in their first paper and show that self-organization occurs among the cooperators, to the exclusion of the cheater. They validate their findings through modeling.

While the referees consider the topic of potential interest for *eLife*, there are two major concerns that need to be clarified before a final decision can be taken. One of these requires additional experiments and the conceptual scope of this paper will depend on the outcome of these experiments. Thus it is currently not possible to state that the paper can be accepted once the experiments are done, but the authors are welcome to resubmit the revised version for new assessment.

The first concern relates to the use of the term “heterotypic cooperation” rather than “mutualism”, which is not simply a semantic question, but has direct relevance for the conceptual framing and the experimental approach to be taken. Due to the current interest in social interactions in microbes, this is a crucial issue for the paper. To qualify for cooperation, you need to show that:

(i) In isolation, cooperators grow faster than cheaters. Here, one probably has to show that R outcompetes C in the presence of G. This crucial point does not seem to be addressed, so the system could simply be a case of negative frequency dependent selection. Even in this case, one could have a case of cooperation, but it still remains to be shown that pure cooperator populations maximize fitness (see e.g., MacLean et al, PLoS Biology, http://dx.doi.org/10.1371/journal.pbio.1000486; please check also definitions of ““cooperator” in Nowack, Science 314, 1560 and Doebeli & Hauert, Ecology Letters 8, 748).

(ii) In any mixture, defectors outcompete co-operators. Probably this is what you refer to when you write that “C had a growth advantage over R”, but this crucial point remains unclear. Instead, you turn off the interaction by providing lysine and adenine in the medium, but this is not the same.

With respect to mutualism, referee 2 sees a need to revisit and properly discuss previous concepts. Currently you posit partner choice/recognition as a possible mechanism favouring mutualism and contrast a spatial environment as an alternative explanation. You state however that the “mechanism is unclear” and that there is a “paradox” in terms of the effect of space. In fact this is an example of “partner fidelity feedback”, a mechanism that is well known in the mutualism literature, so there is no paradox. The essence of partner fidelity feedback is that increasing the fitness of a mutualistic partner itself leads to increased fitness for the other partner. The results can thus be easily explained within this framework: cooperators increase the local fitness of their mutualist partners, thus receiving more cooperation themselves. Empty space is required for cooperators and their partners to grow into in order to self-organise.

To clarify these issues, it will be necessary to repeat the experiments (in both supplemented and un-supplemented media) with monocultures of both the cooperator and cheater strains, showing that growth rate is higher for the cheaters when supplemented, but lower when un-supplemented. Ideally you should be able to show that cheaters should also outcompete cooperators. However, it is acknowledged that this relies on an unstructured environment, so you would need to add a non-spatial treatment. If this is feasible it would be great, but other approaches to solve this issue might also be acceptable.

The second concern is about the details of the modeling. The reason for performing simulations are obscure. Do you indeed imply that simulation results “ensure that no biological mechanism beyond cell growth [...] where required for spatial self organisation”? It seems that modeling efforts would only help to arrive at such statements, but would not be sufficient to “ensure” this. Even more problematic is the fact that the model description is far from sufficient to repeat the simulations. In the equation, D is treated as a spatially varying function, but then referred to as a constant. With two different constant values, there are two different equations, but the position of D in a later equation suggests an unnecessary complication. The text suggests that this equation is solved numerically, but no details are given except the boundary conditions. Instead, you give many simulation results as figures, but it is completely unclear how randomness emerges in the deterministic equations. Is this based on some stochastic initial condition (although you speak of densities)? Or do you use a stochastic algorithm to numerically solve a deterministic equation? In the supplement, you list many parameters (enough to fit all kinds of things), but there is no information on how exactly these enter. The only equation in the paper seems to be almost decoration.

The figures combining simulations and experiments are impressive, but it remains unclear how the various parameters of the model where chosen to generate this similarity (which appears to be difficult to quantify). Moreover, a closer inspection of cluster size and form does suggest that there is some qualitative difference between simulations and experiments. Note that it would not be enough to only clarify the source of parameters – so far the model is not described in a way that one could repeat this study. Please describe it in a way that anyone who reads the paper should be able to come up with his or her own numerical code that leads to the same results.

Finally, the current version of the manuscript is unnecessarily condensed, i.e., Results and Discussion are combined and much important information is relegated to the supplementary material. Given that there are no space constraints, an extended version of the manuscript should be envisaged, including all figures that document results and use supplementary files primarily for additional documentation that would in it self not be required for understanding the work.

[Editors’ note: before acceptance, the following revisions were also requested.]

Thank you for resubmitting your work entitled “Spatial Self-organization Favors Heterotypic Cooperation over Cheating” for further consideration at *eLife*. Your revised article has been favorably evaluated by a Senior editor and a member of the Board of Reviewing Editors. The manuscript has been improved but there are remaining issues that need to be addressed before acceptance, as outlined below.

The comments of referees 2 and 3 are included below. In addition, there was a consultation session with the referees that came to the following conclusion:

The presentation at the places referee 3 has pointed to is a big issue. Taking the perspective of someone reading the paper for the first time, they would quite likely be left confused on many points. A significant improvement in presentation, and addressing the use of non-isogenic lines are the main things the authors need to address.

Reviewer #2:

I have major reservations about the experiments presented in Figure 2 of the revised manuscript, which I feel must be addressed before I can suggest acceptance of the manuscript.

In attempting to demonstrate that a cooperative dilemma occurs the authors pre-adapted both cooperators and cheaters to a lysine-limited environment in order to attempt to remove the confounding effect of adaptation to this environment. While I appreciate that the “adaptive race” could be a large confounding effect, I feel the potential biases introduced are not acceptable. As the strains have been pre-adapted they are likely to no longer be isogenic. As such, all fitness effects observed thereafter could be owing to differing mutations between cooperators and cheaters that have different effects depending on the environment. While much early research on social behaviour in microbes used similarly non-isogenic lines, this has resulted in reduced impact of this work in molecular microbiology. The gold standard now is clearly to use isogenic lines, and I do not feel that I can recommend publication of results using non-isogenic lines in a journal with the aspirations of *eLife*.

The obvious solution to this problem for me would be to remove the experiments presented in Figure 2 from the paper and simply point to the evidence from Shou et al. 2007 and Waite & Shou 2012 supporting the existence of a cooperative dilemma. Alternatively, if the authors could analyse co-cultures on a short enough timescale so that the adaptive race is not a problem this would be preferable, but I don’t know whether this is possible. Regardless I don’t think inclusion of experiments with non-isogenic lines is acceptable for a journal of *eLife’s* standing.

Reviewer #3:

After reading the manuscript carefully, I still do have several concerns. Most importantly, the results are presented in a way that is confusing and most probably not beneficial to the community. For example:

- I have checked with several colleagues and the terms “homotypic cooperation” and “heterotypic cooperation” are not at all familiar terms. From the authors reply, I have not understood why they phrase their work in their own terms.

- “Kin selection can achieve positive assortment” makes no sense.

- “Can increase in frequency if it leads to more offspring that are genetically related to the original co-operator” seems to hold for anything that evolves - this statement seems to be empty unless you indicate that the “more offspring” is not produced by the focal individual.

- “Spatial environment, which allows repeated interactions” is a weird formulation. Usually, these two are treated separately, as repeated interactions allow for sophisticated behaviour conditioned on the past, whereas spatial structure per se does not.

- “Can act as a cheater of the system” is a weird formulation. C cheats G by not providing A, but does it cheat R?

- “break of symmetry from the initial symmetric distribution can be due to stochastic effects such as differences in the initial state of cells” - this seems to be self-contradictory.

- “cells attached to the glass rod” - are you sure all cells equally attach? Is this some biased sample?

I am still not convinced that Figure 3 shows a good agreement between experiment and simulations beyond “red-green domains grow, whereas blue regions remain constrained”. Simulations suggest a more complex micro-structure, which is absent ion the experiment. Figure 3 suggests a much stronger association in the Co&Ch treatment than the experiment. I was also wondering why the authors look at frequencies of the two strains throughout – are absolute population sizes not relevant for the interaction? Why?

In the new, slightly improved model section, the notation should be streamlined and made readable. E.g. K always has two indices, but MM is always the same. This is unusual and not necessary. The nabla operator is not defined and nothing is mentioned on its (probably) discrete spatial version. The metabolite model is basically a time discrete version of a stochastic partial differential equation, is there a reason to use a first order algorithm? Why use two separate grid sizes?

All this suggests a high level of sophistication in terms of numerically formulating the model, but it is not clear that the model results are robust. The dynamics of yeast cells is of course individual based, but some formulations suggest that many choices in this model have been made that are not explicitly described in the text. Thus, the model cannot be understood in detail without looking into the source code, which must be commented in a way that it could be read more easily.

While I find the experimental system and the experimental results interesting, it seems that they follow in a straightforward way from the interactions in the system. Partners and co-operators can coexist and grow to high densities, whereas partners and cheaters cannot, which isolates cheaters.

In summary, while this paper has improved in the revision and while I find the basic results of interest, it is not written in a way that makes it easily accessible. I feel that these results could be presented in a much nicer, less confusing way.

---

## [Author Response]

*“In isolation, cooperators grow faster than cheaters.*”

Do you wonder whether pure cooperators do better than pure cheaters in the presence of the partner? If so, then our answer is yes, pure cooperators indeed do better than pure cheaters. In the absence of adenine and lysine supplements, a coculture of R→A←L and G→L←A can grow to high density, but a coculture of C←L and G→L←A fails to grow (Figure 8, adapted from Shou et al. 2007). We have made this point more explicitly in the main text.

**Author response image 1. fig8:** Adapted from Figures 1 and 6 in (Shou et al. 2007): When partner G→L←A was paired with the cooperator R→A←L, the coculture grew from low to high densities. No persistent growth was observed when partner G→L←A was paired with the cheater C←L.

*“Here, one probably has to show that R outcompetes C in the presence of G.*”

Do you mean C←L outcompetes R→A←L in the presence of G→L←A ? Performing this experiment in a well-mixed environment is not straightforward. The lysine-limited coculture environment strongly selects for adaptive mutants in C←L and R→A←L. Thus, C←L and R→A←L engage in an “adaptive race” (Waite & Shou 2012): if C←L gets the best mutation to grow under lysine limitation, then the coculture is quickly destroyed by cheaters; if R→A←L gets the best adaptive mutation, then the coculture quickly purges cheaters. In this case, C←L outcompetes R→A←L or R→A←L outcompetes C←L depending on which population gets a better mutation, not because of the social interactions. This kind of phenomenon has also been observed for non-engineered cooperating and cheating microbes (Morgan et al. 2012).

In our original manuscript as well as other published work, we have used two approaches to confirm the fitness advantage of C←L over R→A←L while being mindful of the complication introduced by the adaptive race. First, we measured fitness differences between R→A←L and C←L under conditions where adaptive race is minimized. In the presence of abundant lysine to minimize selective pressure, C←L has a 2% fitness advantage over R→A←L in both well-mixed environment (see Figure 1 in (Waite & Shou 2012)) and spatial environment (Figure 1 in the original and revised manuscript). In the absence of lysine so that any adaptive mutants, even if preexisting, could not grow and increase in abundance, R→A←L showed no disadvantage compared to cheaters (Waite & Shou 2012). Thus, C←L has an overall advantage over R→A←L.

Second, we took advantage of an *rsp5* mutation previously found to be highly adaptive for the lysine-requiring cells in a lysine-limited environment (Waite & Shou 2012). When we grew replicate cocultures of *rsp5*
R→A←L, *rsp5*
C←L, and G→L←A in well-mixed liquid environment, we again observed stochastic outcomes with some cocultures dominated by cooperators and other succumbing to cheaters (Figure 1–figure supplement 4 in the original manuscript, now Figure 2 in the revised manuscript). This is indicative of additional rounds of adaptive races between *rsp5*
R→A←L and *rsp5*
C←L toward better adaptation to the lysine-limited coculture environment. We revived from frozen stocks two of these cocultures that exhibited declining R→A←L :C←L (purple). R→A←L :C←L in these revived cocultures continued to decline steadily, suggesting that evolved C←L continued to be more fit than evolved R→A←L by ∼8% (Figure 9). Even though the additional fitness advantage of C←L over R→A←L may not be solely due to adenine overproduction, previous work has used mutations unrelated to social interactions to augment the “cost of cooperation” (Chuang et al. 2010; Gore et al. 2009). We have used these pre-adapted cocultures to show that in a well-mixed liquid environment and in a periodically mixed spatial environment in which spatial self-organization was disrupted, the ratio of cooperator R→A←L to cheater C←L consistently declined (now Figure 2 of the revised manuscript). If we did not perturb the spatial environment, cooperators were favored over cheaters (Figure 2—figure supplement 2, purple).

**Author response image 2. fig9:** Well-mixed evolved cocultures of heterotypic cooperators and cheaters showed a steady change in the R→A←L :C←L ratio. (**A**) To pre-adapt cooperators and cheaters in the lysine-limited coculture environment so that no new mutations of large fitness benefits could quickly arise, we started eight well-mixed replicates (marked by different symbols) consisting of *rsp5*
R→A←L, *rsp5*
C←L, and the ancestral G→L←A (Materials and methods). The *rsp5* mutation was previously found to be highly adaptive for the lysine-requiring cells in a lysine-limited environment (Waite & Shou 2012). The initial stochastic phase at 150 hours is indicative of additional rounds of adaptive races^21^ between *rsp5*
R→A←L and *rsp5*
C←L. After 250 hours, the R→A←L :C←L ratios showed steady trends, suggesting no additional rapid adaptive races. Two of these cultures (purple) were revived from frozen stocks. (**B**) In the two revived cocultures where the evolved populations were denoted with a “ **ʹ** ”, R′→A←L :C′←L continued to decline steadily before being used for experiments in Figure 2—figure supplement 2. Broken axis indicates the period of time elapsed during which a small revived inoculum (30 μl into 200 μl minimal medium and then expanded to 2 ml minimal medium) grew to detectable densities.

In a spatial environment, the same selective pressure of lysine-limitation on R→A←L and C←L cells also exists, but the best mutants are restricted to their original birth locations and thus could not take over the coculture. Hence, deterministic rather than stochastic outcomes were observed when R→A←L, C←L, and G→L←A were grown on a spatial environment either in the absence or the presence of lysine and adenine supplements (Figure 1 in the original and revised manuscript). We have transferred these explanations from supplemental files into the main text.

*“…so the system could simply be a case of negative frequency dependent selection. Even in this case, one could have a case of cooperation, but it still remains to be shown that pure cooperator populations maximize fitness (see e.g., MacLean et al, PLoS Biology,*
http://dx.doi.org/10.1371/journal.pbio.1000486*; please check also definitions of ““cooperator” in Nowack, Science 314, 1560 and Doebeli & Hauert, Ecology Letters 8, 748)*.

*(ii) In any mixture, defectors outcompete co-operators. Probably this is what you refer to when you write that “C had a growth advantage over R”, but this crucial point remains unclear. Instead, you turn off the interaction by providing lysine and adenine in the medium, but this is not the same.*”

The MacLean et al. work used the yeast invertase system, which is known to show negative frequency-dependent selection (i.e., cooperators have an advantage over cheaters when cooperators are rare). In this case, “common goods” – glucose and fructose generated from invertase produced and released from cooperators and not cheaters – are “privatized” to a certain extent so that cooperators have preferential access to the common goods they produced (Gore et al. 2009). Thus, when cooperators are rare, cheaters at high abundance do not have much access to scarce common goods while cooperators do. This advantage of cooperator over cheater when cooperators are rare contributes to the coexistence of cooperators and cheaters. We do not believe that there is negative frequency-dependent selection in our experimental system, for two reasons. First, in a heterotypic cooperative system, an individual needs the common goods supplied by its partner, and therefore privatization of common goods produced by self is futile. Second, as discussed above, we see no signs of negative frequency-dependent selection in our pre-adapted cocultures (Figure 2—figure supplement 2, no trend of stable coexistence of R→A←L and C←L in a well-mixed environment).

Importantly, in our mathematical model, we assumed a fixed cheater advantage over cooperator under all conditions. Experiments and modeling yielded similar results: spatial self-organization favors heterotypic cooperation. Thus, we conclude that negative frequency dependent selection is not a pre-requisite of our conclusions on spatial self-organization favoring heterotypic cooperation over cheating.

*“To clarify these issues, it will be necessary to repeat the experiments (in both supplemented and un-supplemented media) with monocultures of both the cooperator and cheater strains, showing that growth rate is higher for the cheaters when supplemented…*”

As discussed above, we have competed C←L over R→A←L in a well-mixed environment, and we observed an advantage of C←L over R→A←L when lysine is abundant and no disadvantage of C←L over R→A←L when lysine is absent (Waite & Shou 2012). Since in abundant lysine, the fitness advantage of C←L over R→A←L was ∼2%, we cannot simply measure monoculture growth rates. This is because, for example, small differences in growth temperature can give rise to growth rate difference greater than 2%. A competition assay with both R→A←L and C←L ensures that the two competing strains experience the same environment and go through the same amount of dilutions during propagation.

*“but [the growth rate of the cheaters] is lower when un-supplemented.*”

We have also shown that R→A←L + G→L←A but not C←L + G→L←A can grow to high density (Figure 8).

*“However, it is acknowledged that this relies on an unstructured environment, so you would need to add a non-spatial treatment. If this is feasible it would be great, but other approaches to solve this issue might also be acceptable.*”

Using pre-adapted cocultures of R’→A←L, C’←L, and G→L←A in which C’←L was steadily favored over R’→A←L in unstructured (well-mixed or periodically disrupted spatial) environments, we have shown that a spatial environment favors R’→A←L over C’←L (Figure 2 in the revised manuscript).

*“With respect to mutualism, referee 2 sees a need to revisit and properly discuss previous concepts. Currently you posit partner choice/recognition as a possible mechanism favouring mutualism and contrast a spatial environment as an alternative explanation. You state however that the “mechanism is unclear” and that there is a “paradox” in terms of the effect of space. In fact this is an example of “partner fidelity feedback”, a mechanism that is well known in the mutualism literature, so there is no paradox. The essence of partner fidelity feedback is that increasing the fitness of a mutualistic partner itself leads to increased fitness for the other partner. The results can thus be easily explained within this framework: cooperators increase the local fitness of their mutualist partners, thus receiving more cooperation themselves. Empty space is required for cooperators and their partners to grow into in order to self-organise.*”

We have modified our text to link self-organization to partner fidelity feedback.

*“The reason for performing simulations given in line 122 are obscure. Do you indeed imply that simulation results “ensure that no biological mechanism beyond cell growth [...] where required for spatial self organisation”? It seems that modeling efforts would only help to arrive at such statements, but would not be sufficient to “ensure” this.*”

We have changed our text to “We also eliminated the confounding influence of adaptive mutations using mathematical simulations.” Thus, no more use of language such as “ensure.” The original intention was to state that we could use a mathematical model to verify whether a certain set of assumptions is sufficient to generate experimental outcomes.

*“Even more problematic is the fact that the model description is far from sufficient to repeat the simulations. In the equation, D is treated as a spatially varying function, but then referred to as a constant. With two different constant values, there are two different equations, but the position of D in the equation in line 235 suggests an unnecessary complication. The text suggests that this equation is solved numerically, but no details are given except the boundary conditions. Instead, you give many simulation results as figures, but it is e.g. completely unclear how randomness emerges in the deterministic equations. Is this based on some stochastic initial condition (although you speak of densities)? Or do you use a stochastic algorithm to numerically solve a deterministic equation? In the supplement, you list many parameters (enough to fit all kinds of things), but there is no information on how exactly these enter. The only equation in the paper seems to be almost decoration.*”

We have rewritten the modeling aspect of our paper. We have changed diffusion “constant” to diffusion “coefficient” to avoid confusion.

We have added substantially more details on modeling in the main text and in “The diffusion model” section of Materials and methods. In addition, we are enclosing our source code. We also added a sentence addressing stochasticity and determinism All parameters have been experimentally measured except for the three parameters on the carrying-capacity-determining metabolite (such as glucose) required by all three strains (*a*_*G*,_
*K*_*G*,_ and *S*_*G*_)_._ The equation section has also been expanded to be comprehensive.

*“The figures combining simulations and experiments are impressive, but it remains unclear how the various parameters of the model where chosen to generate this similarity (which appears to be difficult to quantify).*”

Most parameters for simulations were experimentally measured, and presumably this is the reason why experiments and simulations shared the same trend. We have added a supplementary figure (Figure 6—figure supplement 1). In this figure, we varied the benefit supply amount and diffusion coefficient in simulations to show that spatial localization of benefits is required to generate results that are similar to experiments.

*“Moreover, a closer inspection of cluster size and form does suggest that there is some qualitative difference between simulations and experiments. Note that it would not be enough to only clarify the source of parameters - so far, the model is not described in a way that one could repeat this study. Please describe it in a way that anyone who reads the paper should be able to come up with his own numerical code that leads to the same results.*”

The difference between simulations and experiments could be due to a variety of reasons. First, our assumption on how cells rearrange to accommodate new cells is clearly simplified. We assumed that with a probability of 0.7, a confined cell will send its new daughter cell straight up and with a probability of 0.3, the new daughter displaces a neighbor cell, causing the displaced neighbor cell and all cells above it to “bulge” up. In reality, the new daughter cell could divide diagonally up. So far, we don’t have ways to experimentally track all possible ways of cell rearrangement and thus our assumption is a simplification of experimental reality. We have now discussed this in Materials and methods. Second, genetically identical cells in the same environment may behave very differently. For example, cells in the top crown of a community are always much brighter than cells down below, even though many cells down below retain the ability to divide (i.e., they are live). We don’t know how these physiological differences may impact nutrient consumption and growth.

*“Finally, the current version of the manuscript is unnecessarily condensed, i.e. Results and Discussion are combined and much important information is relegated to the supplementary material. Given that there are no space constraints, an extended version of the manuscript should be envisaged, including all figures that document results and use supplementary files primarily for additional documentation that would in it self not be required for understanding the work.*”

We have rewritten introduction and added a separate discussion. We have also moved some originally supplemental figures into the main text.

*[Editors’ note: before acceptance, the following revisions were also requested.*]

*“The presentation at the places referee 3 has pointed to is a big issue. Taking the perspective of someone reading the paper for the first time, they would quite likely be left confused on many points. A significant improvement in presentation, and addressing the use of non-isogenic lines are the main things the authors need to address*.

*“Reviewer #2*:

*I have major reservations about the experiments presented in*
Figure 2
*of the revised manuscript, which I feel must be addressed before I can suggest acceptance of the manuscript*.

*In attempting to demonstrate that a cooperative dilemma occurs the authors pre-adapted both cooperators and cheaters to a lysine-limited environment in order to attempt to remove the confounding effect of adaptation to this environment. While I appreciate that the “adaptive race” could be a large confounding effect, I feel the potential biases introduced are not acceptable. As the strains have been pre-adapted they are likely to no longer be isogenic. As such, all fitness effects observed thereafter could be owing to differing mutations between cooperators and cheaters that have different effects depending on the environment. While much early research on social behaviour in microbes used similarly non-isogenic lines, this has resulted in reduced impact of this work in molecular microbiology. The gold standard now is clearly to use isogenic lines, and I do not feel that I can recommend publication of results using non-isogenic lines in a journal with the aspirations of* eLife.

*The obvious solution to this problem for me would be to remove the experiments presented in*
Figure 2
*from the paper and simply point to the evidence from Shou et al. 2007 and Waite & Shou 2012 supporting the existence of a cooperative dilemma. Alternatively, if the authors could analyse co-cultures on a short enough timescale so that the adaptive race is not a problem this would be preferable, but I don’t know whether this is possible. Regardless I don’t think inclusion of experiments with non-isogenic lines is acceptable for a journal of* eLife*’s standing.*”

The cooperating and cheating populations resulting from the pre-adaptation process are not clonal, let alone isogenic. Nonetheless, the growth rate of well-mixed cultures decreased as the ratio of cheaters to cooperators increased, suggesting that cheaters remained cheaters. Top-views of the communities were similar to results from isogenic strains: spatial association of cooperators with the partner and isolation of cheaters. These, and the fact that in nature, cooperators have to face cheaters some of which are of different species and therefore non-isogenic (such as the non-pollinating wasps of fig), and most importantly, the need to show that perturbation of spatial environment causes cheaters to rise in abundance, originally made us to include the experimental result in Figure 2. However, we understand the reviewer’s concern about possible confounding factors due to the usage of non-clonal, non-isogenic populations. We have replaced Figure 2 with simulation data, showing that disruption of spatial organization favors cheaters, which is a very important point we would not want to miss. We have moved the original Figure 2 to supplementary information, and modified the text.

*“Reviewer #3*:

*After reading the manuscript carefully, I still do have several concerns. Most importantly, the results are presented in a way that is confusing and most probably not beneficial to the community. For example*:

*- I have checked with several colleagues and the terms “homotypic cooperation” and “heterotypic cooperation” are not at all familiar terms. From the authors reply, I have not understood why they phrase their work in their own terms.*”

In the literature, mutually beneficial interactions between two species have been referred to as “mutualism” (Foster & Wenseleers 2006), “cooperation” (Harcombe 2010), or “interspecific cooperation” (Sachs et al. 2004). We opted to use “cooperation” rather than “mutualism” to emphasize that the act of helping the partner confers a cost to self. This leads to the social dilemma where cheaters that do not help the partner have fitness advantage over their cooperating counterparts. We believe that using just the term “cooperation” is not adequate, because we want to emphasize that in our system, cooperation involves exchange of distinct benefits between two distinct populations rather than cooperation involving exchanging of identical benefits between identical individuals which the majority of the field has been studying. The term “interspecific cooperation” implies that cooperators are between different species. Even though our yeast strains can be regarded as different “species” because they do not mate with each other, they can also be considered the same species if one uses the divergence in ribosomal RNA sequences to define species (a field standard)! Most importantly, heterotypic cooperation can apply to cooperation between phenotypically differentiated subpopulations, such as between different cell types in a multicellular organism. We have thus chosen to use “homotypic cooperation” and “heterotypic cooperation” to most precisely represent the context of our work. We feel that using precisely defined terms, even if they are unfamiliar, is less confusing than using familiar terms that are not quite appropriate.

*“ “Kin selection can achieve positive assortment” makes no sense.*”

We have modified the paragraph to: “In homotypic cooperation that involves genetic relatives, positive assortment can be achieved through “kin selection”…”

*“ “Can increase in frequency if it leads to more offspring that are genetically related to the original co-operator” seems to hold for anything that evolves - this statement seems to be empty unless you indicate that the “more offspring” is not produced by the focal individual.*”

This sentence, which turned out to be unnecessary, has been deleted.

*“ “Spatial environment, which allows repeated interactions” is a weird formulation. Usually, these two are treated separately, as repeated interactions allow for sophisticated behaviour conditioned on the past, whereas spatial structure per se does not.*”

In a spatial environment, it is more likely (compared to a well-mixed environment) that two neighboring individuals remain neighbors. We have modified the sentence to read: “Spatial environment, which facilitates repeated interactions because two neighboring individuals are likely to remain neighbors, has been shown to promote heterotypic cooperation.”

*“ “Can act as a cheater of the system” is a weird formulation. C cheats G by not providing A, but does it cheat R?*”

For more clarity, we have changed the sentence to:

“These results collectively suggest that C←L acts as a cheater variant of R→A←L.”

*“ “break of symmetry from the initial symmetric distribution can be due to stochastic effects such as differences in the initial state of cells” - this seems to be self-contradictory.*”

We have changed the sentence to:

“Break of symmetry from the initial symmetric spatial distribution can be due to stochastic effects such as differences in the initial amounts of metabolites cells possessed, death of cells, or the random direction of cell division.”

*“ “cells attached to the glass rod” - are you sure all cells equally attach? Is this some biased sample?*”

In pilot experiments, the results were similar when we compared the ratios of samples obtained using the glass rod with the ratios measured for the rest of the sampled community. Thus, based on the pilot experiment and the estimates of ratios from top-view images, we think the sampling bias, even if present, will not affect our conclusions.

*“I am still not convinced that*
Figure 3
*shows a good agreement between experiment and simulations beyond “red-green domains grow, whereas blue regions remain constrained”. Simulations suggest a more complex micro-structure, which is absent ion the experiment.*
Figure 3
*suggests a much stronger association in the Co&Ch treatment than the experiment. I was also wondering why the authors look at frequencies of the two strains throughout - are absolute population sizes not relevant for the interaction? Why?*”

There are differences in the details of patterns even between replicates of simulated or experimental communities. However, the feature that we are focusing on, i.e., more association of cooperators compared to cheaters with the partner, is consistent in all cases. This is the aspect that we are focusing on as self-organization, and it is supported by both simulations and experiments. The microstructure in simulated results that the reviewer is referring to is likely caused by the cell rearrangement assumption in the model. Each new daughter cell in simulations can divide in a random direction, and each division event independently causes rearrangement of some cells in the community. In contrast, in experimental communities, presumably cells rearrange according to the collective pressure exerted by all growing cells. As such, some artifacts, including finer structure at the individual cell level, is expected in the simulation results, which would explain larger values of association index compared to the experiment. Alternatively, some of the finer spatial features in the experimental cross-sections may have been lost in fixing and cryosectioning processes, leading to smaller association indexes when compared to the simulations. Nevertheless, when compared with the competition control, both simulations and experiments clearly show self-organization, which is the main message of our paper.

Since we are focusing on whether cooperator or cheater is favored, we have chosen to focus on the ratio of the two. When cheater frequency is high, population growth is reduced (in blue-dominated region, growth is restricted – see Figure 3 vertical cross-sections). In a strict sense, population size matters. For example, when the total population size and/or density are too low, the community fails to grow even in absence of cheater because the released metabolites are so diluted that cells could not take them up fast enough to remain viable. However, our experiments never used such low population sizes.

*“In the new, slightly improved model section, the notation should be streamlined and made readable. E.g. K always has two indices, but MM is always the same. This is unusual and not necessary. The nabla operator is not defined and nothing is mentioned on its (probably) discrete spatial version. The metabolite model is basically a time discrete version of a stochastic partial differential equation, is there a reason to use a first order algorithm? Why use two separate grid sizes? All this suggests a high level of sophistication in terms of numerically formulating the model, but it is not clear that the model results are robust. The dynamics of yeast cells is of course individual based, but some formulations suggest that many choices in this model have been made that are not explicitly described in the text. Thus, the model cannot be understood in detail without looking into the source code, which must be commented in a way that it could be read more easily.*”

We have deleted the MM. ∇=∂∂xx^+∂∂yy^+∂∂zz^ is the standard vector differential operator.

The following sentence is added:

“∇=∂∂xx^+∂∂yy^+∂∂zz^ is the vector differential operator.”

We explained why we are using two separate spatial domains:

“These grid sizes accommodated different diffusion coefficients in agarose and in community and represented the average distance nutrient molecules diffuse in 3.5 sec.”

Using a first-order algorithm seems to work, since we did not see numerical instability (such as metabolite concentrations going negative). Also, the simulation results qualitatively matched experimental results. For example, growth rates of simulated communities are similar to those of experimental communities.

With regard to the robustness of model results, we have tried altered assumptions and obtained similar results. For example, in Figure 3—figure supplement 1, “Sim w/ delay” incorporated a 60-hour delay in the death of G→L←A and in the consequent release of lysine according to experimental observations. This simulation generated qualitatively similar results as simulations without incorporating the delay. We have also tried to use the same diffusion coefficient for agarose and community, and obtained the same result. We have added:

“When we used the same diffusion coefficient (360 μm^2^/sec and grid size of 50 μm) for agarose and community, similar results were obtained.”

With regard to cell rearrangement, we have added:

“Time-lapse images of the growth of a single fluorescent cell into a microcolony showed that the center of the microcolony became brighter due to multiple cell layers when the microcolony grew to larger than 5-cell radius.”

The details of the diffusion model, including the choice of the discrete version to solve for the diffusion of metabolites, the use of two separate grids, and the details of the rearrangement scheme are discussed thoroughly in our previous *eLife* publication (30). We have added a sentence in the description of the model (Materials and Methods) to make this point clear:

“We provide a summary of the most relevant features of the model below, without repeating the implementation details that can be found in (30).”

We have also added more comments to the source code, and partitioned the code to different sections for improved readability.

*“While I find the experimental system and the experimental results interesting, it seems that they follow in a straightforward way from the interactions in the system. Partners and co-operators can coexist and grow to high densities, whereas partners and cheaters cannot, which isolates cheaters.*”

Many papers, theoretical and experimental, have been devoted to demonstrating that spatial environment stabilizes homotypic cooperation, which seems very straightforward: cooperators cluster with their cooperative offspring and cheaters with their cheating offspring, so of course this favors cooperation, especially if other resources are non-limiting. This straightforwardness did not prevent those reports from having a high impact on how we think, because scientific demonstration rests on different standards from intuition (which is also very important).

One of the most important aspects of our work is perhaps to make the field more aware of how tricky it can be to distinguish partner choice from partner fidelity feedback in natural systems. Our community is a bona-fide partner fidelity feedback system devoid of any possibility for partner recognition, and it looks very much like the situation between legume and cooperating and cheater rhizobia: exclusion of cheaters. Thus, in the absence of direct evidence of recognition, interpreting an “apparent” discrimination of cheaters as “partner choice” can be flawed. We have added this sentence to the Discussion:

“Our work suggests that any argument on partner recognition in a system that is intrinsically spatial…”